# TERRA transcription destabilizes telomere integrity to initiate break-induced replication in human ALT cells

Bruno Silva [1,4], Rajika Arora [1,3,4], Silvia Bione [2] & Claus M. Azzalin [1✉]

Alternative Lengthening of Telomeres (ALT) is a Break-Induced Replication (BIR)-based mechanism elongating telomeres in a subset of human cancer cells. While the notion that spontaneous DNA damage at telomeres is required to initiate ALT, the molecular triggers of this physiological telomere instability are largely unknown. We previously proposed that the telomeric long noncoding RNA TERRA may represent one such trigger; however, given the lack of tools to suppress TERRA transcription in cells, our hypothesis remained speculative. We have developed Transcription Activator-Like Effectors able to rapidly inhibit TERRA transcription from multiple chromosome ends in an ALT cell line. TERRA transcription inhibition decreases marks of DNA replication stress and DNA damage at telomeres and impairs ALT activity and telomere length maintenance. We conclude that TERRA transcription actively destabilizes telomere integrity in ALT cells, thereby triggering BIR and promoting telomere elongation. Our data point to TERRA transcription manipulation as a potentially useful target for therapy.

[1] Instituto de Medicina Molecular João Lobo Antunes (iMM), Faculdade de Medicina da Universidade de Lisboa, Lisbon, Portugal. [2] Computational Biology Unit, Institute of Molecular Genetics Luigi Luca Cavalli-Sforza, National Research Council, Pavia, Italy. [3] Present address: Institute for Molecular Health Sciences, ETHZ, Zürich, Switzerland. [4] These authors contributed equally: Bruno Silva, Rajika Arora. ✉email: cmazzalin@medicina.ulisboa.pt

Transcription of telomeric DNA into the long noncoding RNA TERRA is an evolutionarily conserved feature of eukaryotic cells with linear chromosomes[1]. RNA polymerase II produces TERRA proceeding from subtelomeric regions towards chromosome ends and using the C-rich telomeric strand as a template. As a result, TERRA molecules comprise chromosome-specific subtelomeric sequences followed by a variable number of telomeric UUAGGG repeats[2–4]. TERRA is found either dispersed throughout the nucleoplasm or associated with telomeric chromatin, as well as other genomic loci that contain or not telomeric DNA repeats[4–7]. The molecular mechanisms mediating TERRA retention on chromosomes still need to be fully elucidated; however, the propensity of TERRA to form RNA:DNA hybrids with its template DNA strand (telomeric R-loops or telR-loops)[8–11] and the physical interaction of human TERRA with the shelterin factors TRF1 and TRF2[12,13] suggest that TERRA association with telomeric DNA-containing loci involves RNA–DNA and RNA–protein interactions.

The chromosomal origin of human TERRA is controversial. Using RT-PCR and Illumina sequencing, independent laboratories reported on the existence of TERRA molecules originating from a multitude of chromosome ends[3,14–19]. Consistently, we previously identified CpG dinucleotide-rich tandem repeats of 29 bp displaying promoter activity and located on approximately half of chromosome ends[2]. 29 bp repeats are positioned at variable distances from the first telomeric repeat and their transcriptional activity is repressed by CpG methylation[2]. Moreover, transcription factor binding sites exist on multiple subtelomeres and inactivation of some of them alter TERRA levels in cells[14,20–22]. However, work from the Blasco laboratory, based on re-analysis of TERRA Illumina sequencing data and molecular and cell biological validation experiments, posed that human TERRA is mainly transcribed from one unique locus on the long arm of the chromosome 20 (20q) subtelomere[23,24]. The same group used CRISPR/Cas9 to delete a 8.1 kb fragment from the 20q subtelomere comprising 4 putative promoters in U2OS osteosarcoma cells and isolated several clonal lines (20q-TERRA KO cells). Seemingly supporting the proposed origin of TERRA, 20q-TERRA KO cells displayed substantially diminished total TERRA levels when compared to parental cells[23,24].

TERRA is involved in several telomere-associated processes including telomerase recruitment and regulation, telomeric DNA replication, telomeric heterochromatin establishment, response to DNA damage at telomeres and replicative senescence initiation[1]. Our laboratory and others have implicated TERRA also in telomere elongation in telomerase-negative cancer cells with an activated Alternative Lengthening of Telomeres (ALT) mechanism[8,9,24,25]. ALT is a specialized pathway repairing and thus re-elongating damaged telomeres through Break-Induced Replication (BIR) occurring in the G2 and M phases of the cell cycle and requiring the DNA polymerase delta accessory subunits POLD3 and POLD4[25–29]. Consistent with a function for TERRA in ALT, human ALT cells, including U2OS, are characterized by elevated telomeric transcription and TERRA levels and abundant telR-loops[2,8,15]. Moreover, the RNA:DNA endoribonuclease RNaseH1 and the ATPase/helicase FANCM dismantle telR-loops and FANCM restricts total TERRA levels specifically in ALT cells. Because RNaseH1 and FANCM inactivation increase telomere instability and ALT activity, while their over-expression alleviates ALT[8,9,30,31], we proposed that physiological damage triggered by TERRA/telR-loops at ALT telomeres may provide the substrate for BIR-mediated telomere elongation[8,9,32,33]. However, due to a lack of tools to rapidly suppress TERRA transcription in cells, our hypothesis remained speculative. Further challenging our hypothesis, 20q-TERRA KO cells show increased telomeric localization of the DNA damage factors γH2AX and 53BP1 and

telomeric fusions[24], which has been interpreted as evidence for TERRA capping, rather than destabilizing, telomeres in ALT cells.

## Results

### Development of Transcription Activator-Like Effectors binding to 29 bp repeats.
To assess the short-term impact of TERRA transcription on telomere stability in ALT cells we engineered Transcription Activator-Like Effectors (TALEs)[34] targeting a 20 bp sequence within the 29 bp repeat consensus[2] (herein referred to as T-TALEs; Fig. 1a). Variable numbers of exact 20 bp sequences are found within the last 3 kb of 20 subtelomeres (3p, 5p, 9p, 12p, 16p, 19p, 1q, 2q, 4q, 5q, 6q, 10q, 11q, 13q, 15q, 16q, 19q, 21q, 22q, and Xq/Yq). T-TALEs were C-terminally fused to a strong nuclear localization signal (NLS), four transcription repressor domains of the mSIN3 interaction domain (Enhanced Repressor Domain, SID4X), and a human influenza hemagglutinin (HA) epitope (Fig. 1a). T-TALEs not fused to SID4X were used as controls. Transgenes were cloned downstream of a doxycycline (dox) inducible promoter and stably integrated in U2OS cells expressing the tetracycline repressor protein. Several clonal cell lines were isolated in absence of dox and successively tested for dox-induced T-TALE expression by western blot and indirect immunofluorescence (IF) using anti-HA antibodies. Two independent cell lines for SID4X-fused T-TALEs (sid1 and sid4) and two for unfused T-TALEs (nls1 and nls3) were chosen for further experiments because: (i) transgene expression was almost undetectable in absence of dox; and (ii) dox treatments induced expression of ectopic proteins homogeneously distributed across the cell population, properly localized to the nucleus and at fairly similar levels in the four cell lines (Fig. 1a and Supplementary Fig. 1a–c).

To confirm T-TALE binding specificity, we treated nls3, sid4, and parental U2OS cells with dox for 24 h and performed chromatin immunoprecipitations (ChIPs) with anti-HA antibodies followed by qPCR using oligonucleotide amplifying subtelomeric sequences from several chromosome ends either containing or not 29 bp repeat sequences (29 bp+ and 29 bp−, respectively). DNA in very close proximity of 29 bp repeats on chromosomes 9p, 10q, 15q, 16p, and Xq subtelomeres was enriched in nls3 and sid4 ChIP samples over parental U2OS samples. The enrichment diminished for sequences more distant from the 29 bp repeats on the same subtelomeres (Fig. 1b and Supplementary Table 1). On the contrary, no enrichment was observed for DNA from 29 bp− subtelomeres (10p, 12q, 18p, 20q, and Xp), the centromere of the X chromosome, centromeric alphoid repeats, and the beta Actin or U6 gene loci (Fig. 1b and Supplementary Table 1). This confirms that T-TALEs specifically bind to 29 bp repeat.

### T-TALEs rapidly suppress TERRA transcription from 29 bp repeat-containing chromosome ends.
To test the functionality of T-TALEs, we treated cells with dox for 24 h and performed RT-qPCR to measure TERRA from 29 bp+ (9pXq, 10q, 15q, and 16p) and 29 bp− (10p18p, 12q, 20q, and Xp) subtelomeres. In sid1 and sid4 cells, TERRA from 29 bp+ subtelomeres was greatly reduced while TERRA from 29 bp− was not affected. In nls1 and nls3 cells, no significant change in TERRA levels was observed at any chromosome ends (Fig. 1c). Northern blot hybridization of total RNA with a C-rich telomeric repeat probe did not detect substantial variations across sid and nls cells treated or not with dox for 24 h (Supplementary Fig. 2). Hence, TERRA from 29 bp repeat promoters does not majorly contribute to the cellular pool of UUAGGG repeats. Moreover, because TERRA transcription from 29 bp− chromosome ends is not affected in dox-treated sid cells (Fig. 1c), an immediate cross-talk between the transcriptional state

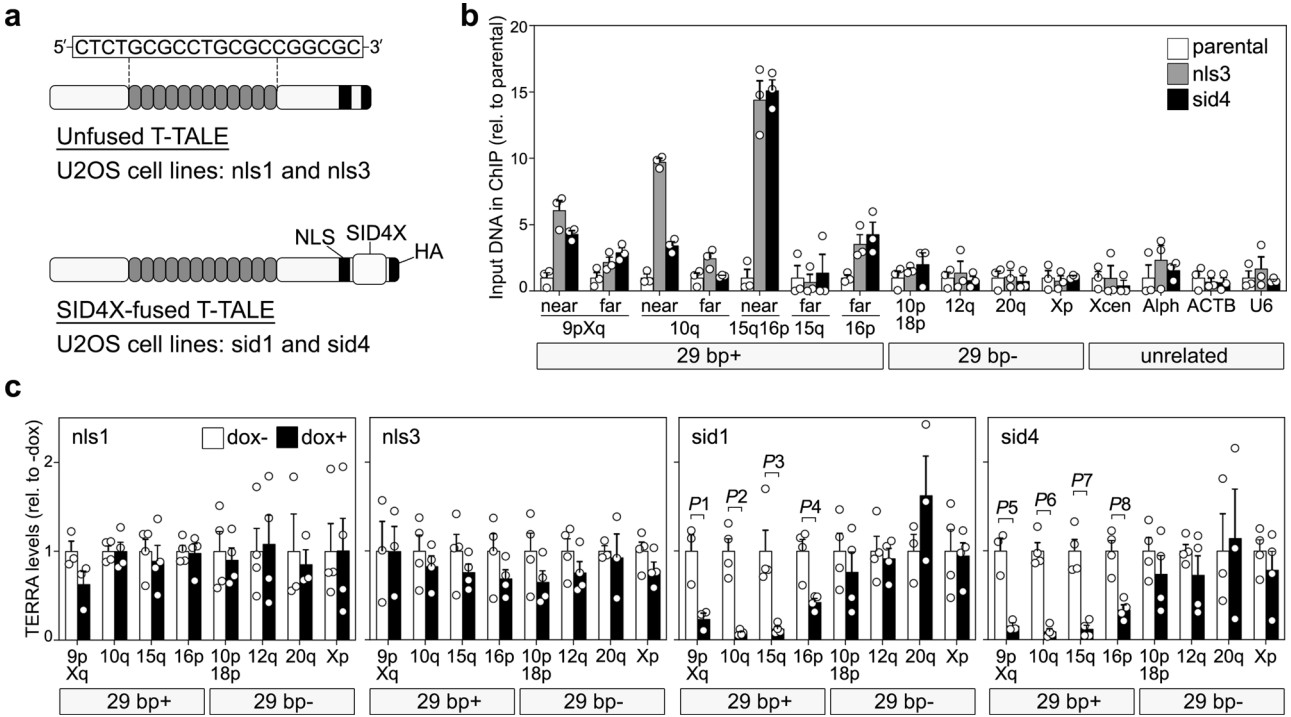

**Fig. 1 Development and validation of T-TALEs. a** Schematic representation of T-TALEs. The RVD domain recognizing the indicated nucleotides within the 29 bp repeat consensus sequence is represented by gray rounded rectangles. NLS: nuclear localization signal; SID4X: four transcription repressor domains of the mSIN3 interaction domain; HA: human influenza hemagglutinin tag. **b** Quantification of anti-HA ChIPs in the indicated cell lines (parental: T-Rex-U2OS) treated with dox for 24 h. QPCRs were performed with oligonucleotides amplifying subtelomeric regions from chromosome ends containing or devoid of 29 bp repeats (29 bp+ and 29 bp−, respectively). For 29 bp+ subtelomeres, two oligonucleotide pairs placed at different distances from the 29 bp array were used and are indicated as near and far. The distances of each amplicon from 29 bp repeats and/or telomeric repeats are shown in Supplementary Table 1. Control qPCR was performed with oligonucleotides amplifying sequences from a unique region of the X chromosome centromere (Xcen), alphoid DNA (Alph), and beta Actin (ACTB) and U6 gene loci. Values are graphed as input DNA found in the corresponding ChIP samples normalized to U2OS parental samples. Bars and error bars are means and SEMs from three independent experiments. Circles are single data points. **c** RT-qPCR quantifications of TERRA transcripts from 29 bp+ and 29 bp− chromosome ends in the indicated cell lines, treated with dox for 24 h or left untreated. Values are graphed normalized to −dox. Bars and error bars are means and SEMs from 3 independent experiments for 9pXq and 20q and from 4 independent experiments for the remaining chromosome ends. Circles are single data points. *P*-values were calculated with a two–tailed Student's *t*-test. *P1* = 0.01886; *P2* = 0.0065; *P3* = 0.0325; *P4* = 0.0059; *P5* = 0.00442; *P6* < 0.0001; *P7* = 0.0006; *P8* = 0.0022. Source data are provided as a Source Data file.

of independent telomeres appears not to exist in U2OS cells. However, a larger number of ends would need to be tested to corroborate this conclusion.

Fluorescence-activated cell sorting (FACS) of propidium iodide (PI)-stained cells did not uncover cell-cycle profile alterations in dox-treated nls and sid cells as compared to untreated controls (Supplementary Fig. 3a, b); this indicates that the TERRA decrease observed in sid1 and sid4 cells is not an indirect consequence of a disturbed cell-cycle progression. Further supporting this notion, in ALT cells, TERRA levels do not diminish when the cell-cycle progresses from S to G2 phases, as typical in telomerase positive cells[4,35]. Hence, our TALE-based system can efficiently and specifically inhibit TERRA transcription from 29 bp promoter repeats.

**TERRA transcription suppression alleviates telomere instability.** To probe the effects of TERRA transcription inhibition on telomere stability, we performed indirect immunofluorescence (IF) using antibodies against the single-stranded DNA binding protein RPA32, RPA32 phosphorylated at serine 33 (pSer33) or γH2AX combined with either telomeric DNA fluorescence in situ hybridization (FISH) or IF against the shelterin component TRF2. RPA32 and pSer33 were used as markers of DNA replication stress, while γH2AX as a broad DNA damage marker.

Dox treatments diminished the telomeric localization of both RPA32 variants in sid1 and sid4 cells (Fig. 2a, b and Supplementary Fig. 4) already at 24 h after drug delivery. A slightly sharper decrease was observed for total RPA32 than for pSer33, suggesting that a fraction of the protein binds to telomeres independently of serine 33 phosphorylation or that the protein undergoes dephosphorylation while still telomere-bound. Similarly, dox treatments diminished the frequencies of γH2AX co-localization with telomeres in sid1 and sid4 cells (Fig. 2a, b, and Supplementary Fig. 4). Dox treatments did not alter co-localization frequencies for any of the tested markers in nls1 and nls3 cells (Fig. 2a, b and Supplementary Fig. 4). Moreover, dox did not affect the total cellular levels of RPA32, pSer33, γH2AX, and TRF2 nor did it impair RPA32 and H2AX phosphorylation when cells were simultaneously treated with the damaging agent camptothecin (Supplementary Fig. 1c). Hence, all changes observed in dox-treated sid1 and sid4 cells do not derive from altered protein cellular levels or from a compromised DNA damage response. Alterations in cell-cycle distribution also cannot account for the observed changes in RPA32, pSer33, and γH2AX at telomeres (Supplementary Fig. 3a, b).

**TERRA transcription suppression inhibits ALT activity.** According to our model, diminished telomere instability should

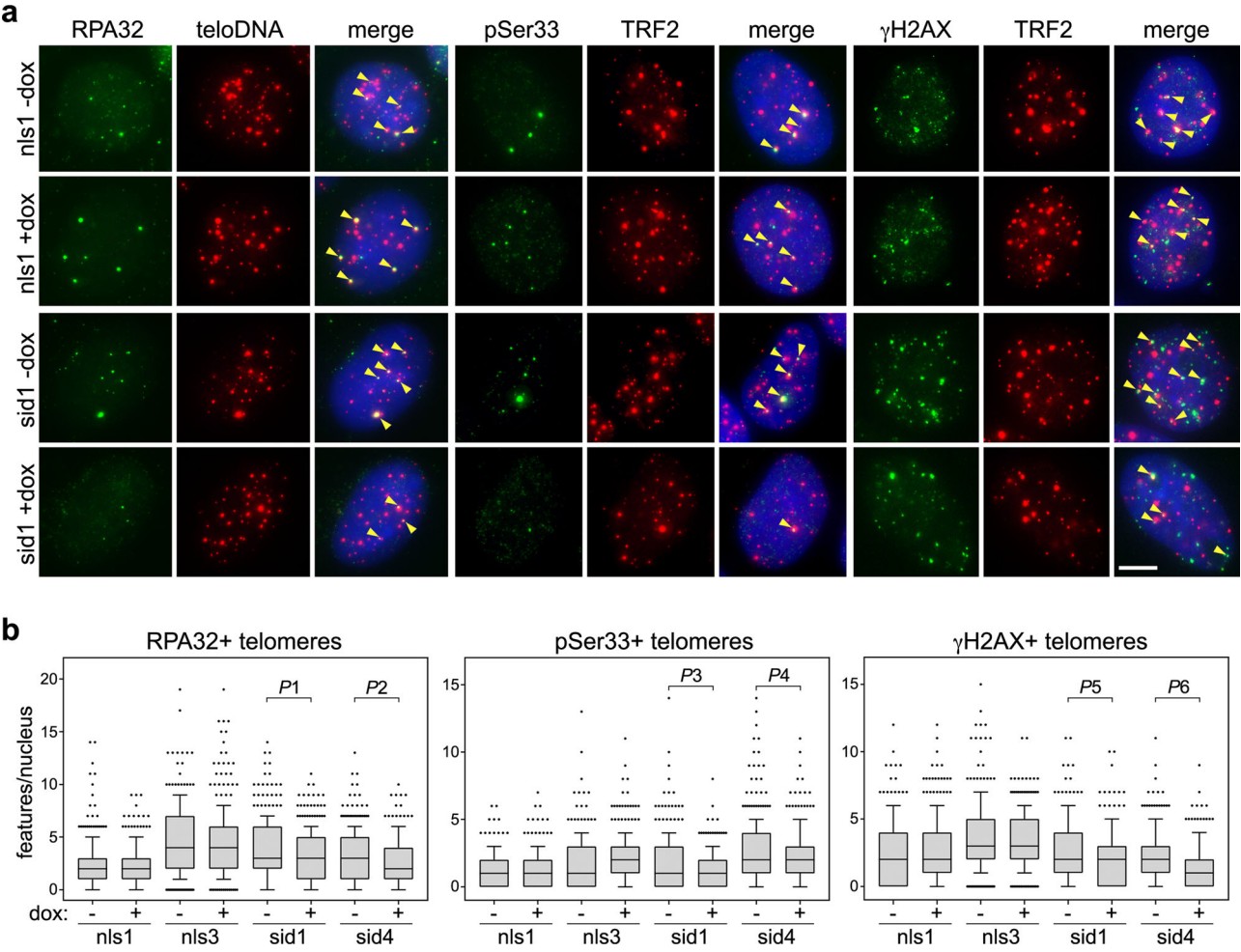

**Fig. 2 TERRA transcription inhibition alleviates telomere instability. a** Examples of RPA32, pSer33, or γH2AX IF (green) combined with telomeric DNA FISH (teloDNA) or TRF2 IF (red). DAPI-stained DNA is in blue. The indicated cell lines were treated with dox for 24 h for RPA32 and pSer33 or 72 h for γH2AX. Arrowheads in the merge panels point to co-localization events. Scale bar: 5 μm. **b** Quantifications of experiments as in **a**. Boxplots represent the medians (middle lines) and the first and third quartiles (boxes), the whiskers extend to the 90th percentile of the values. A total of at least 300 nuclei from three independent experiments were analyzed for each sample. *P*-values were calculated with a two-tailed Mann–Whitney *U* test. P1 = 0.0008; P2 = 0.0010; P3 = 0.0511; P4 = 0.0080; P5 < 0.0001; P6 < 0.0001. Source data are provided as a Source Data file.

weaken ALT activity. To test this, we first quantified ALT-associated PML bodies (APBs) by combining IF with an anti-PML antibody and telomeric DNA FISH. APBs diminished in sid1 and sid4 but not in nls1 and nls3 cells already 24 h after adding dox (Fig. 3a, b and Supplementary Fig. 4). We then synchronized cells at the G1/S transition and let them progress from S-phase to G2 in presence of dox and the Cdk1 inhibitor RO-3306. Cells were pulsed with EdU during the last 2.5 h of treatment and subjected to EdU detection combined with telomeric DNA FISH. Dox did not affect the frequencies of EdU co-localization with telomeric DNA in nls1 and nls3 cells, while it substantially diminished them in sid1 and sid4 cells (Fig. 3a, b and Supplementary Fig. 4). This suggests that dox treatments reduced telomeric BIR in G2 synchronized sid cells. Consistently, as shown by double IF experiments, dox diminished the frequencies of POLD3 co-localization with the shelterin component RAP1 in sid1 and sid4, but not in nls1 and nls3 G2 cells (Fig. 3a, b and Supplementary Fig. 4). Changes in APBs and POLD3 telomeric localization occurred in absence of changes in PML, POLD3, and RAP1 total protein levels (Supplementary Fig. 1b). Moreover, dox treatments did not affect the efficiency of our synchronization protocol (Supplementary Fig. 3a, b). Thus, the

decline in ALT features observed in sid1 and sid4 cells cannot be ascribed to altered protein levels or differences in the fraction of cells in the G2 phase.

As additional markers of ALT activity, we also quantified C-circles, which are circular telomeric DNA molecules with exposed single-stranded C-rich tracts, and telomeric sister chromatid exchanges (TSCEs)[26]. C-circle assays with total genomic DNA did not disclose consistent changes in nls1, sid1, and sid4 cells treated with dox for up to 72 h. In nls3 cells, C-circle levels diminished after 24 h of treatment and started to recover at later time points (Supplementary Fig. 5a, b). Chromosome orientation FISH (CO-FISH) on nls3 and sid4 metaphase chromosomes did not detect significant changes in TSCE frequencies associated with dox treatments in either cell line (Fig. 4a–c). However, we noticed unequal distribution of leading and lagging strand signals at several chromosome ends. Thus, we also quantified the occurrence of sister telomeres with 2 leading and one lagging strand signals (double leading or DLead) and with 2 lagging and one leading strand signals (double lagging or DLagg). DLagg telomeres were not affected by dox treatments in both cell lines, while DLead telomeres were more than halved in sid4 but not nls3 dox-treated cells as compared to untreated controls (Fig. 4a–c).

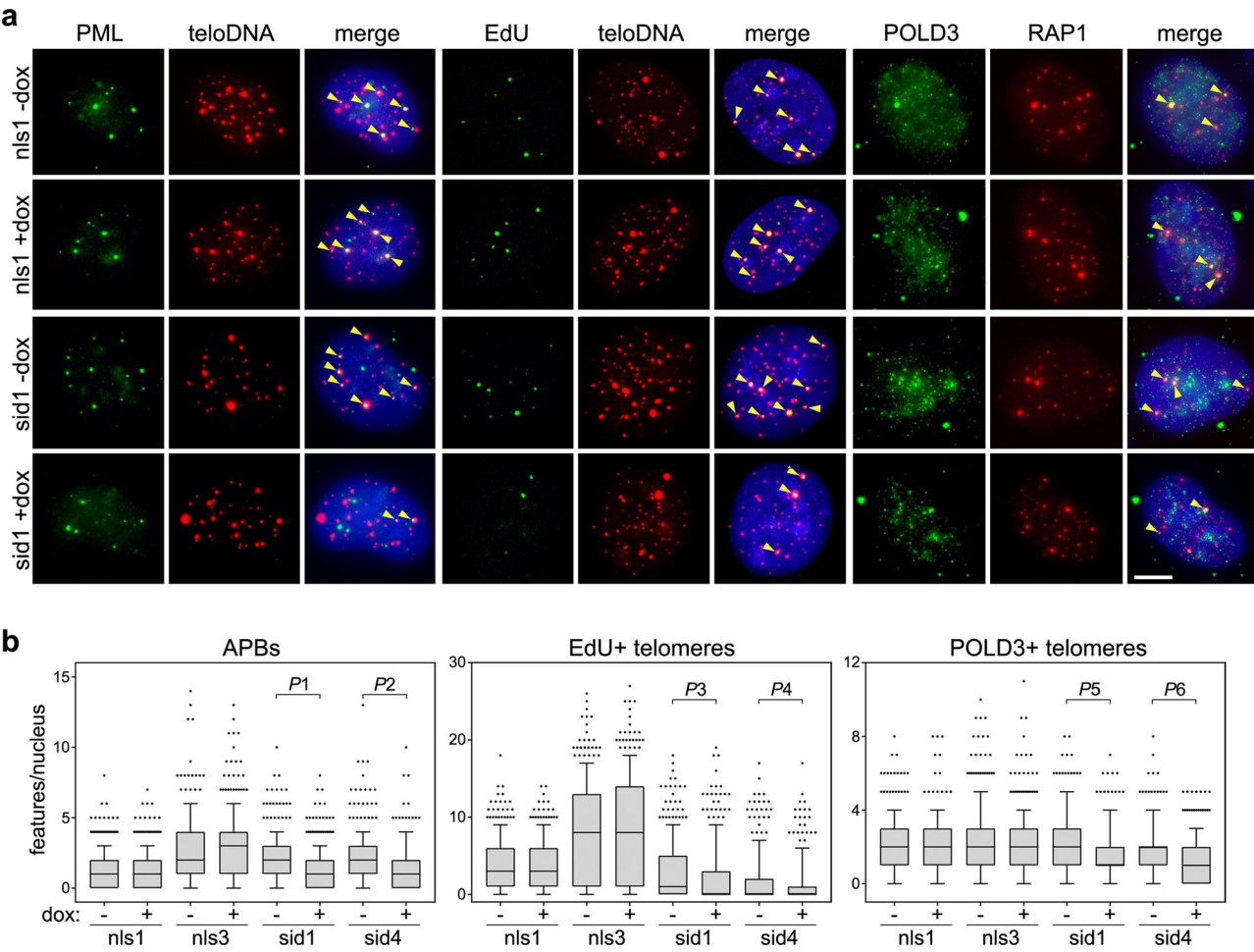

**Fig. 3 TERRA transcription inhibition alleviates ALT activity. a** Left and right panels: examples of PML or POLD3 IF (green) combined with telomeric DNA FISH (teloDNA) or RAP1 IF (red). Middle panels: examples of EdU detection (green) combined with telomeric DNA FISH (red). DAPI-stained DNA is in blue. The indicated cell lines were treated with dox for 24 h for PML or for 24.5 h for POLD3 and EdU (G2 synchronized cells, see methods for details). Arrowheads in the merge panels point to co-localization events. Scale bar: 5 μm. **b** Quantifications of experiments as in **a**. Boxplots represent the medians (middle lines) and the first and third quartiles (boxes), the whiskers extend to the 90th percentile of the values. A total of at least 300 nuclei from three independent experiments were analyzed for each sample. *P*-values were calculated with a two-tailed Mann–Whitney *U* test. *P*1 = 0.0169; *P*2 < 0.0001; *P*3 = 0.0193; *P*4 = 0.0812; *P*5 < 0.0001; *P*6 = 0.0063. Source data are provided as a Source Data file.

**TERRA transcription suppression impairs telomere maintenance.** Alleviation of ALT activity should translate into impaired telomere elongation and progressive loss of telomeric DNA. We treated cells with dox over a prolonged time course and analyzed telomeres by telomeric DNA FISH on metaphase chromosomes. A progressive, statistically significant accumulation of telomere free ends (TFEs) was observed in sid1 and sid4 cells treated with dox for up to 15 and 9 days, respectively (Fig. 5a, b). Conversely, no significant change in TFEs was observed in nls3 cells during the tested time course (Fig. 5a, b). We also inspected the state of telomeres in nls3, sid1, and sid4 cells treated with dox for 15 days by telomere restriction fragment analysis combined with pulse-field gel electrophoresis (PFGE). No major differences were observed in the length of the bulk telomeres (comprised between 10 and 63.5 kb) or in the associated signal intensity when comparing dox-treated and untreated samples (Supplementary Fig. 6). This indicates that the TFEs induced by TERRA transcription inhibition mostly accumulate at shorter telomeres, which are not detected in PFGE. This is consistent with shorter telomeres having higher transcriptional activity[5,16,36] and thus being primarily affected by its inhibition.

We then tested whether crippling the ALT BIR machinery causes the same telomeric defects observed in dox-treated sid1 and sid4 cells. We depleted POLD3 in U2OS cells using two independent siRNAs for 6 days (Supplementary Fig. 1d) and performed metaphase telomeric DNA FISH. POLD3 depletion increased TFE frequencies by almost three-fold as compared to siRNA control-transfected cells (Fig. 5a, b). We conclude that impairing BIR through TERRA transcription inhibition or POLD3 depletion causes substantial loss of telomeric DNA in a relatively short time frame. In agreement with this conclusion, we previously reported that over-expressing RNaseH1 in several ALT cell lines resolves telR-loops and increases TFEs within a period of 6 to 13 days[8].

**TERRA transcription suppression leads to transcriptomic changes.** TERRA depletion in mouse ES cells caused significant changes in gene expression[7]. We, therefore, performed RNA-seq of total RNA prepared from nls3 and sid4 cells treated with dox for 72 h or left untreated. Comparative analysis of dox-treated versus untreated samples identified significant (Benjamini–Hochberg corrected *P* ≤ 0.05) differential expression of 3063 transcripts (2574 protein coding and 489 noncoding transcripts) in sid4 cells; of those transcripts, 1928 were upregulated and 1135 downregulated (Fig. 6a and Supplementary Data 1). In nls3 cells treated with dox, only 15

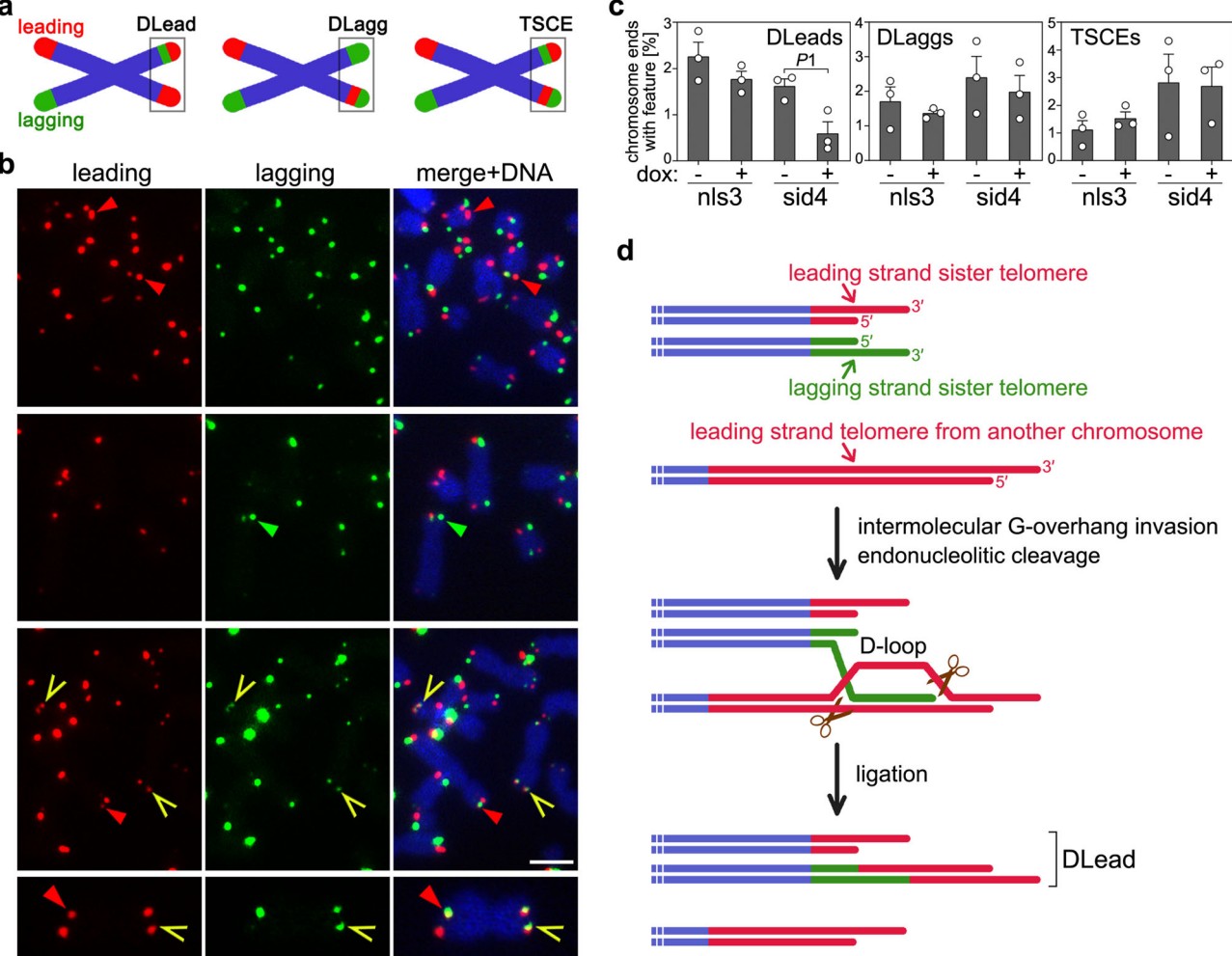

**Fig. 4 TERRA transcription inhibition diminishes the frequencies of DLead chromosome ends. a** Schematic representation of telomeric features scored in CO-FISH experiments. DLead: sister telomeres with 2 leading and one lagging strand signals; DLagg: sister telomeres with 2 lagging and one leading strand signals; TSCEs: telomeric sister chromatid exchanges. **b** Examples of CO-FISH on metaphases from sid4 cells treated with dox for 72 h. Leading and lagging strand telomeres are in red and green, respectively; DAPI-stained DNA is in blue. Red arrowheads point to DLeads, green arrowheads to DLaggs, and yellow arrows to TSCEs. Scale bar: 5 μm. A chromosome with one DLead and one TSCE at its two opposite ends is shown at a 3-fold magnification at the bottom. **c** Quantifications of telomeric features in CO-FISH experiments as in **b**. A total of at least 2538 chromosomes from 3 independent experiments were analyzed for each condition. Bars and error bars are means and SEMs. Circles are single data points. *P*-values were calculated with a two-tailed Student's *t*-test. P1 = 0.0268. **d** Speculative model for DLead generation. Scissors represent structure-specific endonucleases. See the "Discussion" section for details. Source data are provided as a Source Data file.

transcripts (13 protein coding and 2 noncoding) were differentially expressed, 6 being upregulated and 9 downregulated (Supplementary Data 1). Transcript level alterations in sid4 cells were overall moderate, with only 226 and 236 transcripts showing a log2 fold change of at least 1 or −1, respectively (Supplementary Data 1). Hence, TERRA transcription suppression is associated with mild yet significant transcriptomic changes in human ALT cells. Gene Ontology Biological Process (GO-BP) analysis identified alterations of transcripts significantly enriched in 196 GO-BP terms (false discovery rate [FDR] *P* ≤ 0.05) and traceable to 12 main biological processes including nervous system development, cell movement, cell-cycle progression, and cell death (Fig. 6b and Supplementary Data 2). Interestingly, we observed significant upregulation of transcripts involved in vacuole organization and autophagy, and downregulation of transcripts involved in apoptotic cell death (Supplementary Fig. 7 and Supplementary Data 2).

We also noticed a differential expression of transcripts involved in DNA metabolism and identify 11 protein-coding transcripts (downregulated: ORC1, RAD51AP1, BRCA2, GEN1, ORC6,

BRCA1, HROB, BLM, EXO1, FANCD2; upregulated: LIG4) whose altered expression could alleviate telomere instability and ALT activity (Fig. 6a). Despite the differential expression of those transcripts being borderline significant, we quantified the levels of the encoded proteins by western blotting of total proteins from nls and sid cells treated with dox for 0, 24, and 72 h. Although fluctuations in protein levels could be detected for some factors, they likely derived from a general, short-term response to dox, as they were observed in all cell lines and only at 24 h of treatment (Fig. 6c). No specific alteration in protein levels could be detected in sid1 and sid4 cells at any time points (Fig. 6c). This confirms that the effects exerted by SID4X-fused T-TALEs on telomere stability and ALT activity derive from TERRA transcription suppression and not from deregulated off-target transcripts.

## Discussion

We have developed an efficient system to suppress TERRA transcription from several chromosome ends in an ALT cell line.

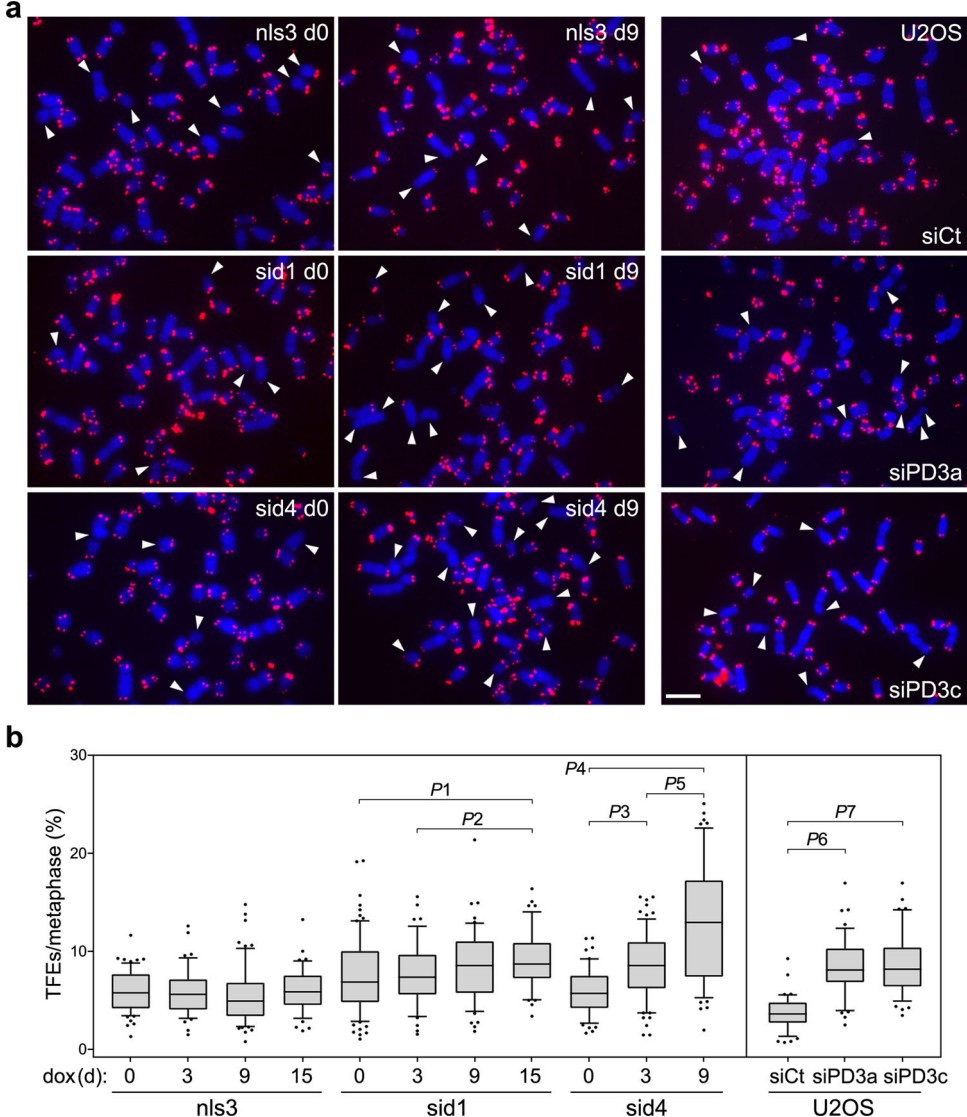

**Fig. 5 TERRA transcription inhibition leads to accumulation of TFEs. a** Examples of telomeric DNA FISH on metaphases from the indicated cell lines. For TERRA transcription inhibition, nls3, sid1, and sid4 cells were treated with dox for up to 15 days (d). For POLD3 depletion, T-REx-U2OS cells were transfected twice three days apart with two independent siRNAs against POLD3 (siPD3a and siPD3c) or with control siRNAs (siCt) and harvested 6 days from the first transfection. Telomeric repeat DNA is in red, DAPI-stained chromosomal DNA in blue. White arrowheads point to chromosome arms with TFEs. Scale bar: 10 μm. **b** Quantifications of experiments as in **a**. Boxplots represent the medians (middle lines) and the first and third quartiles (boxes), the whiskers extend to the 90th percentile of the values. A total of at least 2761 chromosomes from 2 or 3 independent experiments were analyzed for each condition. *P*-values were calculated with a two-tailed Mann–Whitney *U* test. $P1 = 0.0092$; $P2 = 0.0109$; $P3 = < 0.0001$; $P4 < 0.0001$; $P5 < 0.0001$; $P6 < 0.0001$; $P7 < 0.0001$. Source data are provided as a Source Data file.

Importantly, rapid suppression of TERRA transcription across a cell population provides the critical advantage to study the immediate consequences on telomere homeostasis, avoiding secondary effects associated with clonal selection and expansion after TERRA inhibition.

Our findings unmistakably settle that multiple chromosome ends are actively transcribed and further validate that the previously identified 29 bp repeats are functional and physiologically relevant TERRA promoters[2]. The lack of major changes in total UUAGGG levels in dox-treated sid1 and sid4 cells indicates that TERRA transcribed from 29 bp repeat promoters does not majorly contribute to the cellular UUAGGG pool, either because promoters other than the 29 bp repeats have a stronger transcriptional activity or because TERRA from 29 bp repeat promoters has a shorter half-life. While previous data suggest that 20q-TERRA may be the major contributor to the cellular

UUAGGG pool[23,24], the diminished levels of cellular UUAGGG repeats in 20q-TERRA KO cells might also derive from the short telomeres in those cells and/or clonal variability. It would be interesting to systematically deplete TERRA transcripts from different chromosome ends, including 20q, and measure total cellular UUAGGG repeats over a short period of time. A comparison between the stability of TERRA transcripts from different subtelomeres should also help clarify this matter.

Our data establish that TERRA transcription inhibition impairs the accumulation of DNA instability markers (RPA32 and γH2AX) at telomeres, weakens ALT features (APBs and POLD3-dependent synthesis of telomeric DNA in G2 cells), and causes TFE generation. We propose that, in ALT cells, TERRA transcription is a major trigger of replication stress-associated telomere instability and, in turn, BIR-mediated telomere elongation. The only partial decrease in telomere instability and ALT activity

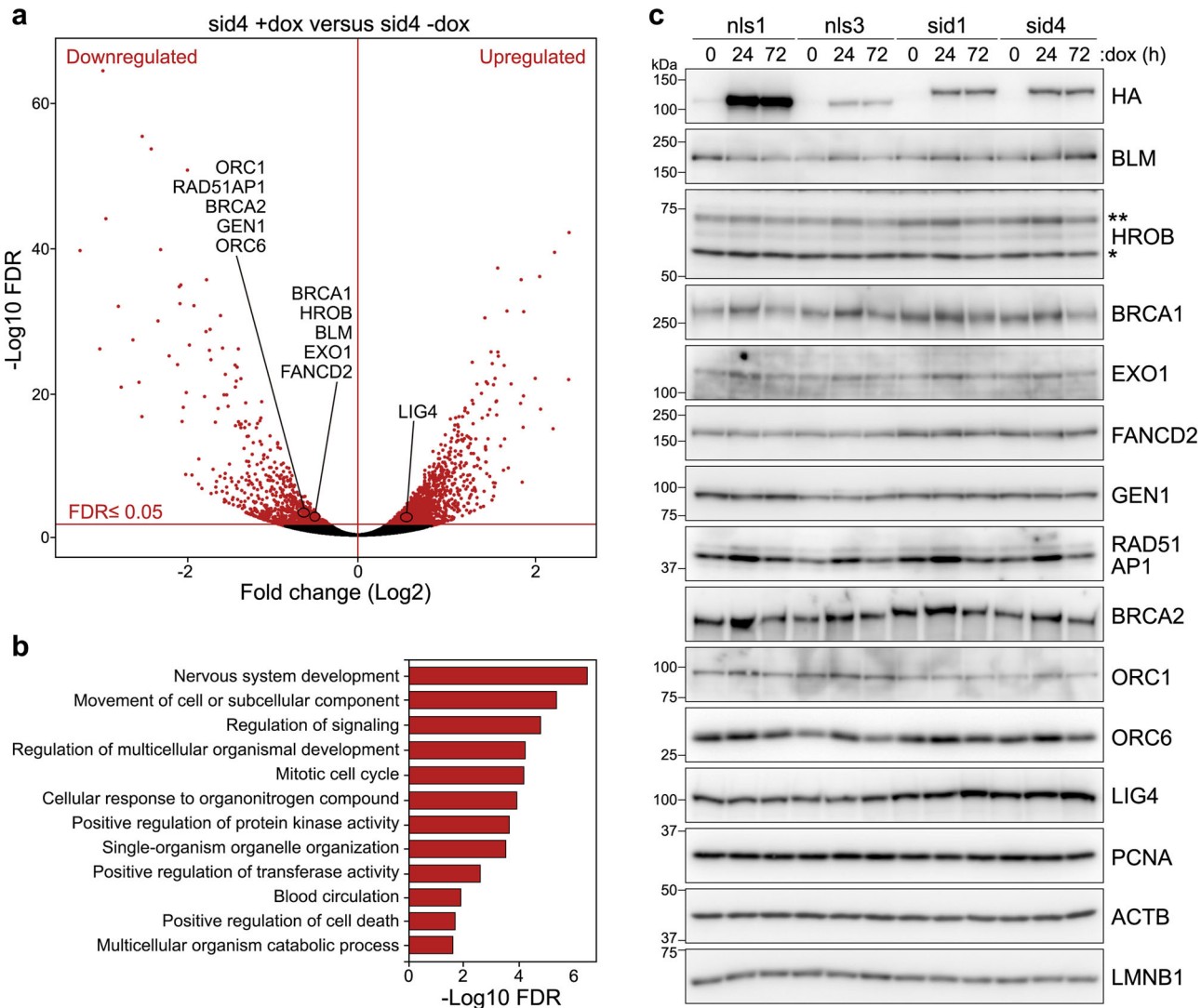

**Fig. 6 TERRA transcription inhibition leads to transcriptomic changes. a** Volcano plot of RNA-seq transcriptome data displaying the pattern of gene expression values in the comparison of sid4 cells treated with dox for 72 h versus untreated sid4 cells. Significantly differentially expressed genes ($P \leq 0.05$ of Benjamini–Hochberg correction of nominal $P$-values obtained from the applied two-sided Wald test) are in red. Genes involved in DNA metabolism are indicated. **b** Bar plot representation of the Gene Ontology analysis of differentially expressed genes identifying significantly enriched biological process functional categories (FDR $P \leq 0.05$). Statistical significance for each term is indicated on the $X$ axis by the −log10 of the FDR $P$-value. **c** Western blot analysis of DNA metabolism factors identified in **a**. The indicated cell lines were treated with dox for 24 or 72 h or left untreated. Images are from different identical membranes. PCNA, Beta Actin (ACTB) and Lamin B1 (LMNB1) serve as loading controls. For HROB, the two asterisks indicate variants 1 and 4, the single asterisk variant 3. Images are representative of experiments repeated at least twice. Source data are provided as a Source Data file.

observed in dox-treated sid1 and sid4 cells is most likely explained by the fact that T-TALEs target a subset of telomeres and/or the existence of additional telomere instability triggers, for example, G-quadruplex structures[25]. Based on previous studies on RNAseH enzymes[8,37] and FANCM[9] and the ability of R-loops to induce DNA instability[38], it seems likely that TERRA transcription causes replication stress by stalling the replication fork through telR-loop formation. It is also possible that telomere instability derives from the collision between TERRA transcription and telomeric replication forks. Importantly, our data challenge the notion that TERRA supports ALT by capping telomeres, as it was previously proposed based on the massive accumulation of telomeric DNA damage in 20q-TERRA KO cells[24]. On the contrary, our studies indicate that the immediate result of TERRA transcription is the active destabilization of ALT telomere integrity to support elongation and cell immortality. Nonetheless, we do not exclude that chromatin-associated (UUAGGG)n

sequences might also be part of the telomeric cap in ALT and non-ALT cells as previously suggested[7,24,39]. Moreover, although our T-TALEs do not affect 20q-TERRA transcription, which is expected because the 20q subtelomere does not contain 29 bp repeats, impaired telomere maintenance is observed both in our system and in 20q-TERRA KO cells; this suggests that many if not all telomeres in a cell have to stay transcriptionally active to ensure fully effective ALT activity.

It is interesting that not all tested ALT features, including C-circles, are affected when TERRA transcription is inhibited. Two alternative BIR pathways have been shown to co-exist in ALT cells, one RAD52-independent and associated with C-circle production, and the other RAD52-dependent and not leading to C-circle production[29]. It is possible that TERRA only supports RAD52-dependent BIR, although this hypothesis is not consistent with the observed C-circle accumulation in RNaseH1- and FANCM-depleted ALT cells[8,9,30,31]. We thus consider that mild

changes in C-circles upon partial TERRA transcription inhibition may fall below the detection limit of our assays. This limitation would not be unprecedented; for example, in a previous report, siRNA-mediated depletion of POLD3 in U2OS cells did not detectably affect C-circle levels while impairing telomeric BIR[27]. Combining our T-TALE system with POLD3, RAD52, FANCM, or RNaseH1 inactivation should elucidate how TERRA transcription and C-circle generation communicate.

TSCEs are also not affected when TERRA transcription is inhibited. This observation was surprising because 20q-TERRA KO cells are characterized by diminished TSCE frequencies[24]. However, a substantial fraction of the overall short telomeres in 20q-TERRA KO cells might have escaped detection in CO-FISH experiments, thus skewing the results and their interpretation. Regardless, our analysis in T-TALE cells revealed that TERRA transcription promotes the formation of rearranged chromosome ends with leading strand replication DNA present at both sisters (DLeads); hence, the mechanisms leading to TSCEs and DLeads are different. Because TERRA transcription promotes telomeric BIR, we interpret DLead structures as the outcome of premature termination of post-replicative BIR events initiating with a lagging strand telomere invading a leading strand one from another chromosome (Fig. 4d). As soon as a D-loop is formed, it could be resolved by structure-specific endonucleases, for example, the SMX complex, which has been previously implicated in ALT[40–44]. Endonucleolytic cleavage followed by end-joining would translocate the distal part of the leading strand telomere onto the lagging strand one, thus generating a DLead structure (Fig. 4d). Supporting this hypothesis, unequal exchanges of sister telomeres were also observed in ALT cells depleted of POLD3 in absence of detectable changes in equal TSCE frequencies[27].

Events where lagging strand telomeres translocate onto leading strand ones, thereby generating DLagg ends, also seem to occur. However, because TERRA transcription inhibition does not affect DLagg frequencies, different molecular triggers appear to act at leading and lagging strand telomeres to initiate BIR. The specificity of TERRA transcription on DLead frequencies can be explained in two alternatives, not mutually exclusive ways. One possibility is that TERRA transcription increases the propensity of lagging strand telomeres to invade other chromosome ends, for example by inducing replication fork stalling and DSBs through telR-loop formation[38]. Because telR-loops, at least in budding yeast, are more abundant at short telomeres[36], this might constitute a regulatory mechanism directing ALT toward the shortest telomeres in the cell. Otherwise, TERRA transcription could prime leading strand telomeres to act as templates for BIR by altering their structure; this also could depend on the formation of telR-loops, as they might shape the double helix into an ideal entry platform for an annealing reaction involving a switch between TERRA and the 3′ end of the acceptor telomere. This second hypothesis is supported by the observation that, in ALT cells, aberrant telR-loop accumulation due to RNAseH1 depletion causes rapid loss of leading strand telomeres[8].

This work also opens new questions that should be addressed in the future. Our experimental settings cannot tell whether TERRA induces telomere instability in *cis* or in *trans*; this is a very relevant question considering the recently reported propensity of (UUAGGG)n RNA sequences produced from a transfected plasmid to be recruited to telomeres through RAD51-mediated R-loop formation[45]. Because T-TALE expression does not alter total cellular UUAGGG levels and RAD51 is dispensable for ALT BIR[27,29], it seems more likely that TERRA transcription regulates telomere stability and ALT activity mainly in *cis*. However, experimental systems able to discriminate between individual chromosome ends, containing or devoid of 29 bp repeats, should be employed to address this question rigorously. It will also be important to explore the connections, suggested by our transcriptomic data, between TERRA transcription and autophagy- or apoptosis-mediated cell death. A deeper characterization of this puzzle piece will tell whether TERRA transcription suppression, for example through chemical inhibition of TERRA promoter activity, could spearhead novel therapeutic strategies for the selective killing of ALT cancer cells.

## Methods

**Plasmid construction**. A repeat-variable di-residue (RVD) domain specifically targeting the 5′-CTCTGCGCCTGCGCCGGCGC-3′ sequence within the 29 bp repeat consensus sequence was designed using TAL Effector Targeter[46]. Variable numbers of the target 20 bp sequence are identified within the most distal 3 kb of 20 subtelomeres according to a complete clone-based assembly of human subtelomeric regions[47]. A 3560 bp DNA fragment corresponding to a full TALE module comprising the designed RVD followed by an SV40 nuclear localization signal and a human influenza hemagglutinin (HA) tag was synthesized at GenScript. The fragment was cloned into a KpnI/ApaI digested pcDNA5-FRT-TO plasmid (ThermoFisher Scientific) downstream of a doxycycline-inducible CMV promoter (unfused T-TALE). The obtained plasmid was digested with ClaI and EcoRV and ligated to a 429 bp fragment comprising the Enhanced Repressor Domain and synthesized at GenScript (SID4X T-TALE). Plasmid sequences are available upon request.

**Cell culture procedures**. T-TALE expressing cells were generated by FRT-mediated integration of unfused T-TALE and SID4X T-TALE plasmids into T-REx™-U2OS cells expressing the TetR protein (ThermoFisher Scientific). Clonal selection was performed by plating cells at low dilution in high glucose DMEM, GlutaMAX (ThermoFisher Scientific) supplemented with 10% tetracycline-free fetal bovine serum (Pan BioTech), 100 U/ml penicillin-streptomycin (Thermo-Fisher Scientific) and 200 μg/ml hygromycin B (VWR). Individual clones were manually picked and expanded in the same medium. For T-TALE induction, 50 ng/ml doxycycline (dox; Sigma-Aldrich) was added to the culture medium devoid of hygromycin B for 24–72 h; for longer induction times, dox was refreshed every 72 h. Mycoplasma contaminations were tested using the LookOut Mycoplasma PCR Detection Kit (Sigma-Aldrich) according to the manufacturer's instructions. When indicated, cells were treated with 1 μM camptothecin (Sigma-Aldrich) for 6 h. For POLD3 depletion, siRNAs were purchased from Integrated DNA Technologies and transfected twice three days apart at 30 nM concentration using the Lipofectamine RNAiMAX reagent (Invitrogen). Target sequences were: siCt (control): 5′-AUACGCGUAUUAUACGCGAUUAAC-3′; siPD3a: 5′-GAAUU GUUAGUAGGUCUAAAC-3′; siPD3c: 5′-CCAAAGCUGCUGCUAAAACCC-3′.

**Fluorescence-activated cell sorting (FACS)**. Cells were trypsinized and pelleted by centrifugation at $500 \times g$ at 4 °C for 5 min. Cell pellets were fixed in 70% ethanol at −20 °C for 30 min and treated with 25 μg/ml RNaseA (Sigma-Aldrich) in 1× PBS at 37 °C for 20 min. Cells were then centrifuged as above and pellets washed in 1× PBS and stained with 20 μg/ml propidium iodide (Sigma-Aldrich) in 1× PBS at 4 °C for 10 min. Flow cytometry was performed on a BD Accuri C6 (BD Biosciences). Data were analyzed using FlowJo software. The utilized gating strategy is shown in Supplementary Fig. 3c.

**Western blotting**. Cells were trypsinized and pelleted by centrifugation at $500 \times g$ at 4 °C for 5 min. Pellets were resuspended in 2× lysis buffer (4% SDS, 20% Glycerol, 120 mM Tris-HCl pH 6.8), boiled at 95 °C for 5 min and centrifuged at $1600 \times g$ at 4 °C for 10 min. Supernatants were recovered and protein concentrations determined by Lowry assay using bovine serum albumin (Sigma-Aldrich) as a standard. 30 μg of proteins were mixed with 0.004% Bromophenol blue and 1% β-Mercaptoethanol (Sigma-Aldrich), incubated at 95 °C for 5 min, separated in polyacrylamide gels, and transferred to nitrocellulose membranes (Maine Manufacturing, LLC) using a Trans-Blot SD Semi-Dry Transfer Cell apparatus (Bio-Rad). The following primary antibodies were used: a rabbit monoclonal anti-HA (Cell Signaling, 3724; 1:1000 dilution), a rabbit polyclonal anti-RAP1 (Bethyl, A300-306A; 1:2000), a mouse monoclonal anti-PCNA (Santa Cruz Biotechnology, sc-56; 1:10000), a rabbit polyclonal anti-TRF2 (Novus Biologicals, NB110-57130; 1:2000), a mouse monoclonal anti-POLD3 (Novus Biologicals, H00010714-M01; 1:500), a mouse monoclonal anti-ACTB (Santa Cruz Biotechnology, sc-47778; 1:2000), a mouse monoclonal anti-PML (Santa Cruz Biotechnology, sc-966; 1:500), a rabbit polyclonal anti-pSer33 (Bethyl, A300-246A; 1:2000), a rabbit polyclonal anti-RPA32 (Bethyl, A300-244A; 1:1000), a rabbit polyclonal anti-LMB1 (GeneTex, GTX103292S; 1:5000), a rabbit polyclonal anti-H3 (Santa Cruz Biotechnology, sc-10809; 1:4000), a mouse monoclonal anti-γH2AX (Millipore, 05-636; 1:2000), a rabbit polyclonal anti-BLM (Bethyl Laboratories, A300-110A; 1:3000), a rabbit polyclonal anti-HROB (Atlas Antibodies, HPA023393; 1:1000), a mouse monoclonal anti-BRCA1 (Santa Cruz Biotechnology, sc-6954; 1:1000), a mouse monoclonal anti-EXO1 (Santa Cruz Biotechnology, sc-56092; 1:500), a mouse monoclonal anti-FANCD2 (Santa Cruz Biotechnology, sc-20022; 1:500), a rabbit

polyclonal anti-GEN1 (Atlas Antibodies, HPA020078, 1:1000), a rabbit polyclonal anti-RAD51AP1 (GeneTex, GTX115455; 1:2000), a mouse monoclonal anti-BRCA2 (Sigma-Aldrich, OP95; 1:2000), a mouse monoclonal anti-ORC1 (Santa Cruz Biotechnology, sc-398734; 1:500), a rat monoclonal anti-ORC6 (Santa Cruz Biotechnology, sc-32735; 1:500), a rabbit monoclonal anti-LIG4 (Abcam, ab193353; 1:2000). Secondary antibodies were HRP-conjugated goat anti-mouse (Bethyl Laboratories, A90-116P; 1:3000), anti-rabbit (Bethyl Laboratories, A120-101P; 1:3000), and anti-rat (Santa Cruz Biotechnology, sc-2006; 1:5000) IgGs. Signals were acquired using an Amersham 680 blot and gel Imager.

**DNA fluorescence in situ hybridization (FISH) and chromosome orientation FISH (CO-FISH).** Metaphase spreads were prepared by incubating cells with 200 ng/ml Colchicine (Sigma-Aldrich) for 5 h. Mitotic cells were harvested by shake-off and incubated in 0.075 M KCl at 37 °C for 10 min. Chromosomes were fixed in ice-cold methanol/acetic acid (3:1) and spread on glass slides. Slides were treated with 20 µg/ml RNaseA (Sigma-Aldrich), in 1× PBS at 37 °C for 1 h, fixed in 4% formaldehyde (Sigma-Aldrich) in 1× PBS for 2 min, and treated with 70 µg/ml pepsin (Sigma-Aldrich) in 2 mM glycine, pH 2 (Sigma-Aldrich) at 37 °C for 5 min. Slides were fixed again with 4% formaldehyde in 1× PBS for 2 min, incubated subsequently in 70%, 90%, and 100% ethanol for 5 min each, and air-dried. A C-rich telomeric PNA probe (5′-Cy3-OO-CCCTAACCCTAACCCTAA-3′; Panagene) diluted in hybridization solution (10 mM Tris-HCl pH 7.2, 70% formamide, 0.5% blocking solution (Roche)) was applied onto the slides followed by incubation at 80 °C for 5 min and at room temperature for 2 h. Slides were washed twice in 10 mM Tris-HCl pH 7.2, 70% formamide, 0.1% BSA and three times in 0.1 M Tris-HCl pH 7.2, 0.15 M NaCl, 0.08% Tween-20 at room temperature for 10 min each. For CO-FISH, cells were incubated with BrdU:BrdC (3:1, final concentration 10 µM; Sigma-Aldrich) for 16 h prior to metaphase preparation as above. Chromosomes were spread on glass slides, treated with RNaseA as above, and incubated with 10 µg/ml Hoechst 33258 (Invitrogen) in 2× SSC for 15 min at room temperature. Slides were exposed to 365-nm ultraviolet light using a Stratagene Stratalinker 1800 UV irradiator set to 5400 J, and incubated with 3 U/µl Exonuclease III (New England Biolabs) at 37 °C for 30 min. Subsequent hybridizations were performed in 30% formamide, 2× SSC for 3 h at room temperature using first a C-rich telomeric LNA probe (5′-6-FAM-CCCTAACCCTAACCCTAA-3′; Exiqon) and then a G-rich telomeric LNA probe (5′-TYE563-TTAGGGTTAGGGTTAGGG; Exiqon). After each hybridization, slides were washed three times in 2× SSC at room temperature for 10 min. Both for FISH and CO-FISH, DNA was counterstained with 100 ng/ml DAPI (Sigma-Aldrich) in 1× PBS, and slides were mounted in Vectashield (Vectorlabs). Images were acquired with a Zeiss Cell Observer equipped with a cooled Axiocam 506 m camera and a ×63/1.4NA oil DIC M27 PlanApo N objective. Image analysis was performed using ImageJ and Photoshop software.

**EdU incorporation and detection at telomeres.** Cells grown on coverslips were incubated in a medium containing 2 mM Thymidine (Sigma-Aldrich) for 21 h before replacement with fresh dox-containing medium. After 4 h, 10 µM RO-3306 (Selleckchem) was added, and 18 h later 10 µM EdU (ThermoFisher Scientific) was added to the culture medium, followed by a 2.5 h incubation. Cells were hybridized as for DNA FISH, washed twice with 1× PBS, and EdU was detected using the Click-iT EdU Alexa Fluor 488 Imaging Kit (ThermoFisher Scientific) according to the manufacturer's instructions. DNA was counterstained with 100 ng/ml DAPI in 1× PBS and coverslips were mounted on slides in Vectashield. Image acquisition and analysis were as for DNA FISH.

**Indirect immunofluorescence (IF).** For HA detection, cells grown on coverslips were incubated in 100% Methanol (Merk) at −20 °C for 15 min. For all other IF experiments, cells were incubated in CSK buffer (100 mM NaCl, 300 mM sucrose, 3 mM MgCl₂, 0.5% Triton X-100, 10 mM PIPES pH 7) for 7 min on ice, fixed with 4% formaldehyde (Sigma-Aldrich) in 1× PBS for 10 min and permeabilized again with CSK buffer for 5 min. Fixed cells were incubated in blocking solution (0.5% BSA, 0.1% Tween-20 in 1× PBS) for 1 h followed by incubation in blocking solution containing primary antibodies for 1 h, three washes with 0.1% Tween-20 in 1× PBS for 10 min each, and incubation with secondary antibodies diluted in blocking solution for 50 min. For combined IF and DNA FISH, cells were again fixed with 4% formaldehyde in 1× PBS for 10 min, washed three times with 1× PBS, incubated in 10 mM Tris-HCl pH 7.2 for 5 min and then denatured and hybridized with a PNA probe (5′-AF568-OO-CCCTAACCCTAACCCTAA-3′; Panagene) as for DNA FISH. DNA was counterstained with 100 ng/ml DAPI in 1× PBS or in 0.1 M Tris-HCl pH 7.2, 0.15 M NaCl, 0.08% Tween-20. Coverslips were mounted on slides in Vectashield. The following primary antibodies were used: a rabbit monoclonal anti-HA (Cell Signaling, 3724; 1:1000 dilution), a rabbit polyclonal anti-RAP1 (Bethyl, A300-306A; 1:2000), a mouse monoclonal anti-TRF2 (Millipore, 05-521; 1:2000), a mouse monoclonal anti-POLD3 (Novus Biologicals, H00010714-M01; 1:500), a mouse monoclonal anti-PML (Santa Cruz Biotechnology, sc-966; 1:500), a rabbit polyclonal anti-pSer33 (Bethyl, A300-246A; 1:2000), a rabbit polyclonal anti-RPA32 (Bethyl, A300-244A; 1:1000), a mouse monoclonal anti-γH2AX (Millipore, 05-636; 1:2000). Secondary antibodies were Alexa Fluor 568-conjugated donkey anti-rabbit IgGs (ThermoFisher Scientific,

A10042; 1:1000) and Alexa Fluor 488-conjugated donkey anti-mouse IgGs (ThermoFisher Scientific, A21202; 1.1000). Image acquisition and analysis were as for DNA FISH.

**Chromatin immunoprecipitation (ChIP).** Cells were harvested by trypsinization, centrifuged at 500×g at 4 °C for 5 min, and resuspended in 1% formaldehyde (Sigma-Aldrich) for 20 min at room temperature, followed by quenching with 125 mM glycine (VWR) for 5 min. Cross-linked cells were centrifuged as above and pellets were resuspended in ChIP lysis buffer (1% SDS, 10 mM EDTA, 50 mM Tris-HCl pH 8), sonicated using a Bioruptor apparatus (Diagenode) and centrifuged at 16,000×g for 10 min at 4 °C. 1 mg of lysate was diluted in ChIP dilution buffer (150 mM NaCl, 20 mM Tris-HCl pH 8, 1% Triton X-100, 2 mM EDTA) and incubated with 2 µg of a rabbit monoclonal anti-HA antibody (Cell Signaling, 3724) for 2 h at room temperature. Immunocomplexes were isolated by incubation with Protein A/G PLUS-Agarose beads (Santa Cruz Biotechnology) at 4 °C overnight on a rotating wheel. Beads were washed 4 times in ChIP wash buffer (150 mM NaCl, 20 mM Tris-HCl pH 8, 1% Triton X-100, 0.1% SDS, 2 mM EDTA) and once in ChIP final wash buffer (500 mM NaCl, 20 mM Tris-HCl pH 8, 1% Triton X-100, 0.1% SDS, 2 mM EDTA). Beads were incubated in ChIP elution buffer (1% SDS, 100 mM NaHCO₃) containing 40 µg/ml RNaseA (Sigma-Aldrich) for 1 h at 37 °C and DNA was extracted using the Wizard SV gel and PCR cleanup system (Promega). Input and immunoprecipitated DNA was subjected to quantitative PCR using the oligonucleotides shown in Supplementary Table 1. QPCRs were performed using the iTaq Universal SYBR Green Supermix (Bio-Rad) on a Rotor-Gene Q (Qiagen) instrument with a 2-step program (45 cycles of denaturation at 95 °C for 15 s, annealing and extension at 60 °C for 30 s). Data analysis was performed using the Rotor-Gene 6000 Series Software 1.7.

**RNA preparation and analysis.** Total RNA was isolated using the TRIzol reagent (ThermoFisher Scientific) followed by chloroform extraction and treated three times with 3.5 U of DNaseI (Qiagen) for 45 min at room temperature. For RT-qPCR, 5 µg of RNA were reverse transcribed with 0.5 µM TeloR and 0.05 µM ActinR oligonucleotides (Supplementary Table 1) and Superscript III (ThermoFisher Scientific) according to the manufacturer's instructions. Quantitative PCRs were performed and analyzed for ChIP using the oligonucleotides shown in Supplementary Table 1. Actin values were used as normalizers. For northern blotting, 15 µg of RNA were separated in 1.2% agarose gels containing 0.7% formaldehyde and transferred onto nylon membranes. When indicated, gels were incubated with 50 mM NaOH, 1.5 M NaCl for 10 min at room temperature prior to transfer. To detect total TERRA (UUAGGG pool), RNA was hybridized at 55 °C overnight with a double-stranded telomeric probe (Telo2 probe) radioactively labeled using Klenow fragment (New England Biolabs) and [α-³²P]dCTP. Post-hybridization washes were twice in 2× SSC, 0.2% SDS for 20 min and once in 0.2× SSC, 0.2% SDS for 30 min at 55 °C. After the radioactive signal acquisition, membranes were stripped and re-hybridized at 50 °C overnight with the Actin_2 oligonucleotide (Supplementary Table 1) radioactively labeled using T4 Polynucleotide Kinase (New England Biolabs) and [γ-³²P]ATP. Post-hybridization washes were twice in 2× SSC, 0.2% SDS for 20 min and once in 1× SSC, 0.2% SDS for 30 min at 50 °C. Radioactive signals were detected using a Typhoon FLA 9000 imager (GE Healthcare) and quantified using ImageJ software.

**Genomic DNA preparation and analysis.** Genomic DNA was isolated by phenol:chloroform extraction and treatment with 40 µg/ml RNaseA (Sigma-Aldrich), followed by ethanol precipitation. Reconstituted DNA was digested with HinfI and RsaI (New England Biolabs) and again purified by phenol:chloroform extraction. For PFGE, 4 µg of digested DNA were separated in a 1% agarose gel at 15 °C for 21 h (6 V/cm, switch time 5 s, included angle 120°) using a CHEF-DRIII system (Bio-Rad). Gels were dried and hybridized with a radioactively labeled Telo2 probe for northern blotting. For C-circle assays, 500 ng of digested DNA were incubated with 7.5 U of phi29 DNA polymerase (New England Biolabs) in presence of dATP, dTTP, and dGTP (1 mM each) at 30 °C for 8 h, followed by heat-inactivation at 65 °C for 20 min. Amplification products were dot-blotted onto nylon membranes (GE Healthcare) and hybridized with a radioactively labeled Telo2 probe as for northern blotting. Radioactive signals were detected using a Typhoon FLA 9000 imager (GE Healthcare) and quantified using ImageJ software.

**Statistical analysis.** For direct comparison of two groups, we employed a paired two-tailed Student's t-test using Microsoft Excel or a nonparametric two-tailed Mann–Whitney U test using GraphPad Prism. P-values are indicated in figure legends.

**RNA-seq.** Triplicate cultures of sid4 and nls3 cells were treated with dox for 3 days or left untreated. Total RNA extracted as above was subjected to rRNA removal and sequencing using a HiSeq 4000 Illumina sequencer at Novogene Co., Ltd. Approximately 50 million reads per replicate were obtained applying a paired-end protocol (2 × 150 bp). Raw reads were subjected to standard quality control procedures with the FastQC software (v.0.11.8; https://www.bioinformatics.babraham.ac.uk/projects/fastqc/) and aligned to the human genome reference sequence (NCBI38/hg38) with the RNA STAR alignment software (v.2.7.7a)[48]. Genes were

annotated with respect to the GENCODE basic annotation v.36 and quantified with the featureCounts tool (v.2.0.1)[49]. Differential gene expression analysis was performed with DESeq2 (v. 2.11.40.6)[50] and differentially expressed genes were selected applying a statistical threshold of 0.05 of the Benjamini–Hochberg corrected $P$-value. The functional enrichment analyses of differentially expressed genes were performed using the Database for Annotation, Visualization and Integrated Discovery (DAVID, v.6.8)[51]. Volcano and bar plots were generated with the R package ggplot2[52], bubble plots were generated with the R package GOplot[53].

**Reporting summary**. Further information on research design is available in the Nature Research Reporting Summary linked to this article.

## Data availability
All raw RNA sequencing data are publicly available through the NCBI Sequence Read Archive (BioProject ID: PRJNA699729). All other relevant data supporting the findings of this study are available within the paper and its Supplementary Information files or upon request. Source data are provided with this paper.

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

## Acknowledgements

We thank Ana Margarida Figueira and Harry Wischnewski for help with RNA preparation and analysis and with C-circle assays, and the Bioimaging and Flow Cytometry facilities of iMM for services. We thank Ludovica Celli for technical support in the preparation of figures for RNA-seq data. This work was supported by LISBOA-01-0145-FEDER-0 29352, project co-funded by FEDER, through POR Lisboa 2020–Programa Operacional Regional de Lisboa and Fundação para a Ciência e a Tecnologia, the European Molecular Biology Organization (IG3576), the Fundação para a Ciência e a Tecnologia (IF/01269/2015; PTDC/MED-ONC/28282/2017) and the European Union's Horizon 2020 Research and Innovation Programme (GA No. 857119–RiboMed).

## Author contributions

B.S., R.A., and C.M.A. conceived the project, designed and performed the experiments, analyzed the data, and wrote the manuscript. S.B. analyzed the RNA-seq data and helped write the manuscript.

## Competing interests

The authors declare no competing interests.
