## [Peer Review File · Nature Communications]

REVIEWER COMMENTS

Reviewer #1 (Remarks to the Author):

In this manuscript, Silva and colleagues developed a novel approach to inhibit TERRA transcription from multiple chromosomes in an ALT cell line. Using this approach, the authors show that inhibition of TERRA transcription is sufficient to impair ALT activity and relieve DNA replication stress at telomeres. Based on this work the author suggests that TERRA transcription induces replication stress at telomeres and initiates BIR to support telomere elongation. The role of TERRA at telomeres has been controversial in part due to the lack of efficient tools to efficiently downregulate its levels. Previous work aimed at addressing the role of TERRA in ALT cells relied on the clonal expansion of cells with a large deletion in the subtelomeric region of 20q. In contrast, the Azzalin group developed an inducible system to downregulate the expression of TERRA, thus avoiding potential issues arising from the analysis of single-cell clones. This is a solid manuscript that provides critical insight into the role of TERRA in ALT-positive cells. All the data presented are of the highest quality and the conclusions are based on strong data. This is an important piece of work of relevance to the telomere biology community.

I have some minor details that, if addressed, could strengthen the manuscript:

- 1_Are the global levels of TERRA levels diminished upon the induction of T-TALEs? The authors show reduced levels of TERRA transcripts from the specific chromosome ends targeted by T-TALEN expression. It would be relevant to know what is the contribution of these transcripts to the global pool of TERRA.
- 2_The authors suggest that suppression of TERRA inhibits ALT activity. Corroborating this notion, the authors show a reduction in APBs and DNA synthesis at telomeres (Fig. 3) as well as the appearance of telomere-free ends (Fig 5). It would be interesting to know whether the reduction of telomere length occurs across all chromosomes (e.g. by TRF analysis) or whether the effect is localized to a few chromosome ends. In the latter case, is the effect confined to those chromosome ends that experience loss of TERRA expression?
- 3-There is a typo in figure 3a, one of the IF is for POLD3 rather than RAP1.

Reviewer #2 (Remarks to the Author):

This study sets out to address the important and unresolved question of whether human telomeric RNA transcripts (TERRA) contribute to the establishment of the ALT pathway for telomerase-independent telomere maintenance in cancer cells, or whether they might act to suppress it instead.

To address this question the authors establish a very elegant system to control TERRA transcription at a large subset of telomeres. Evidence is presented that, fulfilling the design intent, regulated expression of a TALE-SID fusion that targets 29 bp sites found at nearly half of the human telomeres leads to strong down-regulation of TERRA levels, in at least a subset of these telomeres. The effect is specific for 29 bp-containing telomeres and for presence of the repressive SID domain.

It is shown that expression of TALE-SID leads to diminished telomeric RPA, RPA-pS33, and γ H2AX foci, suggesting that TERRA contributes to elicit a DNA damage signal at telomeres.

To assess the hypothesis that DDR engagement contributes to stimulate ALT at telomeres, the effect of suppressing TERRA levels on a key ALT marker, ALT-associated PML bodies (APBs), is monitored. APBs are indeed found at lower levels in TALE-SID expressing cells. Similarly, DNA synthesis at telomeres in G2, and POLD3 telomeric localisation, were found to be decreased upon TERRA suppression. This is consistent with the view that TERRA contributes to the ALT pathway.

In contrast, two other classic features of ALT, telomeric C-circles and Telomeric Sister Chromatid Exchanges (T-SCEs) are not affected by TERRA manipulations in this system. Instead, in the course of assessing T-SCE levels, reduced levels of unequal exchanges are found at leading

telomeres upon TERRA suppression.

Although the results are somewhat mixed, on the whole the study provides evidence that TERRA might promote ALT activity.

To further test this, the authors go on and check telomere maintenance upon prolonged suppression of TERRA, as inhibition of ALT would be expected to lead to progressive telomere loss. Upon inspection, telomere-free ends (TEFs) are indeed found to increase in TERRA-suppressed cells. Although this result might seem to corroborate the authors' interpretation that TERRA is required for ALT-based telomere maintenance, I am puzzled by the relative rapidity of the effect. This would seem to better fit a pattern of DNA replication problems within the bulk of the telomere which seem to be exacerbated by TERRA loss, and hence would be consistent with the alternative interpretation that TERRA contributes to telomere protection by facilitating replication. I find the interpretation of this experiment more complicated than the authors suggest, and potentially consistent with the opposite view that they are trying to disprove.

Overall I find that this manuscript is highly interesting and of high quality. It introduces a system that should prove useful in future studies on the role of TERRA in ALT. The results presented are not entirely straightforward, particularly regarding the last experiment described above. Does inhibition of ALT in different settings (other than by regulating TERRA transcription) lead to increased TEFs? Would PLD3 or ATRX inhibition lead to similar results in the same timeframe? I think this would be an important experiment to do. If conclusive data exist in the literature to this regard they should be discussed.

In addition, I find that the potential impact of the work is diminished by the (conventional) approach chosen to detect many of the readouts at telomeres in these cells, which is based in immunofluorescence. This approach does not allow to distinguish among individual telomeres (some of which contain 29 bp sites and some which do not), and therefore is expected to dampen any effect, as the authors recognise, of inhibiting TERRA at 29 bp subtelomeres specifically. Given that ChIP was successfully carried out for the T-TALE constructs here, I think that similar ChIP approaches for EdU, RPA, POLD3 etc would be likely to yield better data, that would strengthen the authors' conclusions about a (partial?) requirement for TERRA in the ALT process.

Minor points:

In the ChIP experiment in Fig. 1b it should be specified in the legend how far the primers are to the 29 bp sites (N and F situations); distance should also be made clear for the '29 bp -' samples. I find it odd that the analysis was done for such a small subset of subtelomeres (at least for the N sites). Doing a few additional PCRs seems trivial compared to the extensive work carried out to establish the lines.

Along these same lines, I find it odd that the same telomeres for which there is ChIP data (Fig. 1b) are not exactly the same as those for which data is presented about TERRA levels (Fig. 1c). There are no ChIP data for 16p. It would seem reasonable to assess T-TALE binding and TERRA levels at the same telomeres, to better establish a cause-effect relationship. If there are technical reasons for not doing so, it should be said, otherwise one is left with the doubt that the data might not be fully representative. I am also puzzled as to why one sample id 9pXq in panel c but Xq in panel b: there were specific primers for ChIP but not RT-PCR?

Cell cycle data should be presented to show the G2 state of the cells analysed in Fig. 3 for EdU incorporation. It is not entirely clear that the data in Fig. S2b represent the time of EdU pulsing.

Reviewer #3 (Remarks to the Author):

Silva et al. engineer a new tuneable system to regulate TERRA transcription in a doxycycline-dependent manner in human cells. By this means, they address the effect of TERRA downregulation on the ALT phenotype in U2OS cells.

Upon doxycycline treatment, the TALEs (Transcription Activator-Like Effectors), are able to target the 29 bp repeat consensus present in 20 subtelomeres and influence the expression of TERRA when fused with the enhanced repressor domain SID4X. The authors demonstrate the efficiency and specificity of TALEs by measuring its binding to telomeric and non-telomeric loci, in both presence and absence of the SID4X domain and in +/- doxycycline conditions. In agreement with the expected, TERRA expression is significantly reduced only at the subtelomeric loci containing the 29 bp repeat consensus, when the SID4X domain is fused with TALEs and doxycycline is present.

Following their previous work (Arora et al., 2014; Silva et al., 2019) regarding telomeric R-loops-associated replication stress as a trigger for ALT, the current manuscript aims at further confirming whether TERRA transcription challenges telomere replication and favour homology-directed repair (HDR). To test that, the authors shut down TERRA expression by TALEs and assess typical ALT features. Successfully, markers of telomere fragility (RPA32, pSer33, γ H2AX), telomere synthesis in G2, APBs formation and ALT-specific break-induced replication intermediates significantly diminish compared to controls not treated with doxycycline. The authors confirm this effect is not due to an alteration of the cell cycle progression nor to aberrant protein synthesis upon TALEs expression.

Overall, the observations disclosed here are perfectly in line with the idea that TERRA RNA-DNA hybrids are required in a precise balance for ALT to take place. Lowering their levels by inhibiting TERRA transcription alleviates genome instability and disrupts the HDR mechanism on which ALT is based.

The manuscript is written in a very clear way and the general concept unfolds concisely and easily to understand. It's very well placed as a follow-up story to Arora et al., 2014 and Silva et al., 2019 and provides empirical proof to what is commonly speculated about the replication-TERRA transcription conflict as an ALT inducer. Furthermore, the manuscript is important for defining the still controversial, yet relevant, features of TERRA biology in humans. By this, I refer in particular to the notion regarding the multi-chromosomal origin of the transcript.

I feel that this manuscript is very important in understanding how TERRA contributes to telomere maintenance and can be potentially targeted for therapy. I have one major concern that I think should be addressed before publication at Nat. Comm.

1) In Figure 1, it is shown that TALEs specifically target subtelomeres containing the 29 bp repeat consensus and not other subtelomeres nor non-telomeric loci. More importantly, TERRA expression is downregulated exclusively at the aforementioned telomeres. However, it is not shown whether expression of TALEs affects the expression of other genes that might have an effect on ALT telomeres i.e. recombination factors, DNA polymerase delta subunits etc. I would recommend an RNA seq experiment to ensure that known factors for ALT maintenance are not affected (e.g. FANCM).....or at the least western blots for as many known ALT maintenance factors as possible.

Point-by-point response to the Reviewers' comments.

We would like to thank the three Reviewers for their very thorough and constrictive criticisms, which we have addressed by performing new experiments and/or in the text. We provide below detailed answers to the Reviewers' comments and highlighted all changes in blue throughout the text of the revised manuscript.

Reviewer #1 (Remarks to the Author):

In this manuscript, Silva and colleagues developed a novel approach to inhibit TERRA transcription from multiple chromosomes an ALT cell line. Using this approach, the authors show that inhibition of TERRA transcription is sufficient to impair ALT activity and relieve DNA replication stress at telomeres. Based on this work the author suggests that TERRA transcription induces replication stress at telomeres and initiates BIR to supporting telomere elongation. The role of TERRA at telomeres has been controversial in part due to the lack of efficient tools to efficiently downregulate its levels. Previous work aimed at addressing the role of TERRA in ALT cells relied on the clonal expansion of cells with a large deletion in the subtelomeric region of 20q. In contrast, the Azzalin group developed an inducible system to downregulate the expression of TERRA, thus avoiding potential issues arising from the analysis of single-cell clones. This is a solid manuscript that provides critical insight into the role of TERRA in ALT-positive cells. All the data presented are of the highest quality and the conclusions are based on strong data. This is an important piece of work of relevance to the telomere biology community.

We very much appreciate the supportive comments of this Reviewer and we thank you for recognizing that our data provide critical insight into how TERRA transcription supports ALT.

I have some minor details that, if addressed, could strengthen the manuscript:

1_ Are the global levels of TERRA levels diminished upon the induction of T-TALEs? The authors show reduced levels of TERRA transcripts from the specific chromosome ends targeted by T-TALEN expression. It would be relevant to know what is the contribution of these transcripts to the global pool of TERRA.

We have performed northern blot hybridization of total RNA using a C-rich telomeric repeat probe. RNA was extracted from nls1, nls3, sid1 and sid4 cells treated with dox for 24 hours (or left untreated), a time point when T-TALEs clearly diminish the levels of TERRA from 29 bp repeat promoters in sid cells (Fig. 1c). We could not detect any substantial variations in UUAGGG signals across sid and nls samples (data are shown in Fig. S2). We thus conclude that TERRA from 29 bp repeat promoters does not majorly contribute to the cellular pool of UUAGGG repeats because promoters other than the 29 bp repeats have a stronger transcriptional activity or because TERRA from 29 bp repeat promoters has shorter half-life (lanes 131-134 and 271-282).

2_ The authors suggest that suppression of TERRA inhibits ALT activity. Corroborating this notion, the authors show a reduction in APBs and DNA synthesis at telomeres (Fig. 3) as well as the appearance of telomere-free ends (Fig 5). It would be interesting to know whether the reduction of telomere length occurs across all chromosomes (e.g. by TRF analysis) or whether the effect is localized to a few chromosomes ends. In the latter case, is the effect confined to those chromosome ends that experience loss of TERRA expression?

We have performed telomere restriction fragment analysis combined with pulse field gel electrophoresis (PFGE) using total genomic DNA from nls3, sid1 and sid4 cells treated with

dox for 15 days (or left untreated). No major differences were observed in the length of the bulk telomeres (comprised between 10 and 63.5 kb) or in the associated signal intensity when comparing dox-treated and untreated samples (data are shown in Fig. S6). In our view, these observations indicate that TFEs induced by TERRA transcription inhibition mostly accumulate at shorter telomeres, which are not detected in PFGE. Supporting this interpretation, shorter telomeres are characterized by higher transcriptional activity¹⁻³ and are thus expected to be primarily affected by TERRA transcription inhibition. Data are discussed in the Results section (lanes 211-219)

We have also tried to answer whether TERRA transcription inhibition specifically leads to TFE accumulation at chromosome ends containing 29 bp repeat promoter sequences. We have performed DNA FISH using a combination of probes detecting telomeric repeats and the 10q subtelomere using metaphases from nls3 and sid4 cells treated with dox for 15 days (or left untreated). As shown in Fig. R1, we uncovered accumulation of TFEs at 10q chromosome ends in dox-treated sid4 cells. This shows that TERRA transcription inhibition can at least induce TFEs at one 29 bp+ chromosome end (i.e. *in cis*). However, similar experiments using probes for other subtelomeres (29 bp+ or 29 bp-) proved to be technically challenging and ultimately did not produce robust results. We have decided not to include these data in the manuscript because of their preliminary nature and because, in their absence, the main message of the manuscript remains unchanged. We now highlight this limitation in the Discussion by stating that our experimental settings cannot tell whether TERRA induces telomere instability *in cis* or *in trans* (lanes 354-358). Notably, because T-TALE expression does not alter total cellular UUAGGG levels, it seems more likely that TERRA transcription regulates telomere stability and ALT activity mainly *in cis* (this is also mentioned in the Discussion, lanes 358-363). We plan to establish experimental systems similar to the ones in Fig. R1 and applied to a multitude of chromosome ends containing or not the 29 bp repeat promoters to address this question in follow-up studies.

Figure R1. (a) Examples DNA FISH analysis on metaphase spreads from nls3 and sid4 cells treated with dox for 15 days. Telomeric repeats (shown in red) were detected using a C-rich telomeric PNA, 10q subtelomeres (in green) using a probe generated by nick translation of BAC DNA from clone RP11-108K14 (CHORI); DAPI-stained DNA is in blue. Boxed chromosomes are enlarged on the right of each panel to facilitate visualization. Arrowheads point to TFEs. (b) Quantifications of TFEs at 10q subtelomeres (% of total 10q ends) in the indicated cells lines treated as in a. Bars and error bars are from 3 independent experiments. *P* values were calculated with a two-tailed Student's *t*-test. ***P* < 0.005, ****P* < 0.001.

3_ There is a typo in figure 3a, one of the IF is for POLD3 rather than RAP1.

Thanks for noticing this, we have corrected the typo.

Reviewer #2 (Remarks to the Author):

This study sets out to address the important and unresolved question of whether human telomeric RNA transcripts (TERRA) contribute to the establishment of the ALT pathway for telomerase-independent telomere maintenance in cancer cells, or whether they might act to suppress it instead. To address this question the authors establish a very elegant system to control TERRA transcription at a large subset of telomeres. Evidence is presented that, fulfilling the design intent, regulated expression of a TALE-SID fusion that targets 29 bp sites found at nearly half of the human telomeres leads to strong down-regulation of TERRA levels, in at least a subset of these telomeres. The effect is specific for 29 bp-containing telomeres and for presence of the repressive SID domain.

It is shown that expression of TALE-SID leads to diminished telomeric RPA, RPA-pS33, and γ H2AX foci, suggesting that TERRA contributes to elicit a DNA damage signal at telomeres. To assess the hypothesis that DDR engagement contributes to stimulate ALT at telomeres, the effect of suppressing TERRA levels on a key ALT marker, ALT-associated PML bodies (APBs), is monitored. APBs are indeed found at lower levels in TALE-SID expressing cells. Similarly, DNA synthesis at telomeres in G2, and POLD3 telomeric localisation, were found to be decreased upon TERRA suppression. This is consistent with the view that TERRA contributes to the ALT pathway. In contrast, two other classic features of ALT, telomeric C-circles and Telomeric Sister Chromatid Exchanges (T-SCEs) are not affected by TERRA manipulations in this system. Instead, in the course of assessing T-SCE levels, reduced levels of unequal exchanges are found at leading telomeres upon TERRA suppression.

Although the results are somewhat mixed, on the whole the study provides evidence that TERRA might promote ALT activity. To further test this, the authors go on and check telomere maintenance upon prolonged suppression of TERRA, as inhibition of ALT would be expected to lead to progressive telomere loss. Upon inspection, telomere-free ends (TEFs) are indeed found to increase in TERRA-suppressed cells. Although this result might seem to corroborate the authors' interpretation that TERRA is required for ALT-based telomere maintenance, I am puzzled by the relative rapidity of the effect. This would seem to better fit a pattern of DNA replication problems within the bulk of the telomere which seem to be exacerbated by TERRA loss, and hence would be consistent with the alternative interpretation that TERRA contributes to telomere protection by facilitating replication. I find the interpretation of this experiment more complicated than the authors suggest, and potentially consistent with the opposite view that they are trying to disprove. Overall, I find that this manuscript is highly interesting and of high quality. It introduces a system that should prove useful in future studies on the role of TERRA in ALT. The results presented are not entirely straightforward, particularly regarding the last experiment described above. Does inhibition of ALT in different settings (other than by regulating TERRA transcription) lead to increased TEFs? Would PLD3 or ATRX inhibition lead to similar results in the same timeframe? I think this would be an important experiment to do. If conclusive data exist in the literature to this regard they should be discussed.

First of all, we would like to thank this Reviewer for his/her thorough evaluation of our work and constructive criticisms. We are glad to see that this Reviewer considers our T-TALE system elegant, as we believe that it allows to answer important questions that have remained open until now. We are also glad that this Reviewer finds the work highly interesting and of high quality.

We agree that TFEs seem to appear relatively fast. However, we previously showed that dissolution of telomeric R-loops obtained through RNaseH1 over-expression progressively increases TFE incidence in several ALT cell lines within a period of 6 to 13 days⁴; this is consistent with the results obtained in sid1 and sid4 cells. We have now added this mention in the text (lanes 226-228). Additionally, as suggested by this Reviewer, we siRNA depleted POLD3 in U2OS cells for 6 days and performed metaphase telomeric DNA FISH. POLD3 depletion increased TFE frequencies by almost three folds as compared to siRNA control-

transfected cells (data shown in Fig. 5a and b and discussed in the text, lanes 220-226). Overall, these observations strongly support the view that impairing the BIR ALT machinery in different ways (POLD3 depletion, RNaseH1 overexpression or TERRA transcription inhibition) causes substantial loss of telomeric DNA in a relatively short time frame. Based on the newly performed TRF analysis (please see our answer to Reviewer 1's point 2), we believe that TFEs induced by TERRA transcription inhibition mostly accumulate at shorter telomeres, which can explain the relatively rapid manifestation of the phenotype.

We would also like to add that siRNA-mediated depletion of POLD3 in ALT cells was previously reported not to affect C-circle levels and TSCE (equal exchanges) frequencies, while it diminished unequal exchanges of sister telomeres⁵. These results are reminiscent of what we observed in sid cells treated with dox, further strengthening our suggestion that POLD3 and TERRA transcription genetically interact. This is now mentioned in the text (lanes 334-336).

Finally, we would like to emphasize that our goal was not to disprove the previous work from the Blasco laboratory. However, that 20q-TERRA was proposed to be the main if not unique TERRA locus had to be directly considered in our work. This is why we extensively discussed the Blasco work and tried to find explanations for the apparent discrepancies. However, our main conclusion was that work from independent laboratories, including the work on 20q, and our current data imply that many if not all telomeres have to be kept transcriptionally active for ALT to occur efficiently (lanes 304-306). We have toned down our statements on 20q-TERRA in several parts of the Discussion in order to direct the reader's attention to the results obtained with our T-TALEs and not to the differences (which are anyway explainable) between our work and the previous work on 20q-TERRA.

In addition, I find that the potential impact of the work is diminished by the (conventional) approach chosen to detect many of the readouts at telomeres in these cells, which is based in immunofluorescence. This approach does not allow to distinguish among individual telomeres (some of which contain 29 bp sites and some which do not), and therefore is expected to dampen any effect, as the authors recognise, of inhibiting TERRA at 29 bp subtelomeres specifically. Given that ChIP was successfully carried out for the T-TALE constructs here, I think that similar ChIP approaches for EdU, RPA, POLD3 etc would be likely to yield better data, that would strengthen the authors' conclusions about a (partial?) requirement for TERRA in the ALT process.

We have performed anti- γ H2AX and -RPA32 ChIPs in sid1 cells treated with dox for 24 hours (or left untreated), followed by qPCR analysis using primer pairs specific for 10q, 15q (29 bp+) and 12q (29 bp-) subtelomeres. As shown in Figure R2, we were unable to detect specific changes at 29 bp+ subtelomeres; rather, we observed a generalized, slight increase in immunoprecipitated DNA for all tested subtelomeres in cells treated with dox. This is likely due to a general response to drug treatment, which confounds the interpretation of the results. In addition, while ChIPs are performed on chromatin that has been sonicated to produce short DNA fragments of about 100 bps, the damage induced by TERRA transcription is expected to accumulate throughout the telomeric repeat tract and thus possibly far enough from the subtelomere to escape detection in ChIPs combined with qPCRs using subtelomeric oligonucleotides.

As also explained in our answer to Reviewer's 1 point 2, we now clearly acknowledge in the last paragraph of the Discussion (lanes 354-363) that our experimental approach cannot tell apart the effects exerted by the SID4X T-TALEs on 29 bp repeat-containing telomeres and telomeres devoid of those promoters (*in cis* versus *in trans* effects). However, we remain convinced that our data clearly establish that TERRA is a major initiator of ALT BIR.

Figure R2. Quantification of anti- γ H2AX and -RPA32 ChIPs in *sid1* cells treated with dox for 24 hours or left untreated. QPCRs were performed with oligonucleotides amplifying subtelomeric regions from chromosome ends containing (10q and 15q) or devoid (12q) of 29 bp repeats. Values are graphed as input DNA found in the corresponding ChIP samples and expressed as fold change over untreated (-dox) samples. Bars are averages of technical triplicates from one experiment.

Minor points:

In the ChIP experiment in Fig. 1b it should be specified in the legend how far the primers are to the 29 bp sites (N and F situations); distance should also be made clear for the '29 bp -' samples.

Due to the length of the text associated with this information, we have added it in Supplementary Table S1 and referred to the Table in the legend of Fig. 1. For the 29 bp+ chromosome ends, we have indicated the distance of each amplicon from the upstream 29 bp repeat region and from the downstream telomeric tract. For the 29 bp- chromosome ends we have indicated the distance of each amplicon from the downstream telomeric tract.

I find it odd that the analysis was done for such a small subset of subtelomeres (at least for the N sites). Doing a few additional PCRs seems trivial compared to the extensive work carried out to establish the lines. Along these same lines, I find it odd that the same telomeres for which there is ChIP data (Fig. 1b) are not exactly the same as those for which data is presented about TERRA levels (Fig. 1c). There are no ChIP data for 16p. It would seem reasonable to assess T-TALE binding and TERRA levels at the same telomeres, to better establish a cause-effect relationship. If there are technical reasons for not doing so, it should be said, otherwise one is left with the doubt that the data might not be fully representative. I am also puzzled as to why one sample is 9pXq in panel c but Xq in panel b: there were specific primers for ChIP but not RT-PCR?

We have repeated the anti-HA ChIP experiments and completed the analysis following this Reviewer's suggestion. For consistency with the RT-qPCR results, we have added new ChIP qPCRs for 16p far, 10p18q and 20q. Please note that some primer pairs recognize sequences from two chromosome ends due to a high degree of identity between the two loci (9pXq near, 15q16p near, 10p18q). Moreover, the previous labeling XqN and XqF was incorrect, as the used oligo pairs pick up sequences present on both Xq and 9p subtelomeres. We have now repeated the PCRs with the '9pXq far' primer pair for consistency with the RT-qPCR data (same oligo pair) and corrected the labeling mistake. Thank you for spotting this inconsistency. Please also note that we have repeated RT-qPCR quantifications for 9pXq and 20q (Fig. 1c), for which we presented values from two independent experiments in the first version of the manuscript (3 independent experiment in the revised version).

Cell cycle data should be presented to show the G2 state of the cells analysed in Fig. 3 for EdU incorporation. It is not entirely clear that the data in Fig. S2b represent the time of EdU pulsing.

The PI staining data shown in the original Fig. S2 (now Fig. S3) correspond indeed to the exact times of harvesting of cells for the analysis of RPA32, pSer33, γ H2AX (Fig. 2), EdU and POLD3 (Fig. 3). The lower panels are from cells that have been treated with RO-3306 and EdU following the outline explained in the Methods section. We have now clarified this in the legend of Fig. S3 and added '+EdU' in the figure itself.

Reviewer #3 (Remarks to the Author):

Silva et al. engineer a new tuneable system to regulate TERRA transcription in a doxycycline-dependent manner in human cells. By this means, they address the effect of TERRA downregulation on the ALT phenotype in U2OS cells.

Upon doxycycline treatment, the TALEs (Transcription Activator-Like Effectors), are able to target the 29 bp repeat consensus present in 20 subtelomeres and influence the expression of TERRA when fused with the enhanced repressor domain SID4X. The authors demonstrate the efficiency and specificity of TALEs by measuring its binding to telomeric and non-telomeric loci, in both presence and absence of the SID4X domain and in +/- doxycycline conditions. In agreement with the expected, TERRA expression is significantly reduced only at the subtelomeric loci containing the 29 bp repeat consensus, when the SID4X domain is fused with TALEs and doxycycline is present.

Following their previous work (Arora et al., 2014; Silva et al., 2019) regarding telomeric R-loops-associated replication stress as a trigger for ALT, the current manuscript aims at further confirming whether TERRA transcription challenges telomere replication and favor homology-directed repair (HDR). To test that, the authors shut down TERRA expression by TALEs and assess typical ALT features. Successfully, markers of telomere fragility (RPA32, pSer33, γ H2AX), telomere synthesis in G2, APBs formation and ALT-specific break-induced replication intermediates significantly diminish compared to controls not treated with doxycycline. The authors confirm this effect is not due to an alteration of the cell cycle progression nor to aberrant protein synthesis upon TALEs expression.

Overall, the observations disclosed here are perfectly in line with the idea that TERRA RNA-DNA hybrids are required in a precise balance for ALT to take place. Lowering their levels by inhibiting TERRA transcription alleviates genome instability and disrupts the HDR mechanism on which ALT is based.

The manuscript is written in a very clear way and the general concept unfolds concisely and easily to understand. It's very well placed as a follow-up story to Arora et al., 2014 and Silva et al., 2019 and provides empirical proof to what is commonly speculated about the replication-TERRA transcription conflict as an ALT inducer. Furthermore, the manuscript is important for defining the still controversial, yet relevant, features of TERRA biology in humans. By this, I refer in particular to the notion regarding the multi-chromosomal origin of the transcript.

I feel that this manuscript is very important in understanding how TERRA contributes to telomere maintenance and can be potentially targeted for therapy.

We very much appreciate the supportive comments of this Reviewer and we thank you for recognizing the relevance of our work and its timely nature. We are also glad that this Reviewer recognizes that our work not only helps clarify how TERRA contributes to telomere maintenance in ALT cells but also opens the way for testing novel therapeutic options targeting TERRA transcription.

I have one major concern that I think should be addressed before publication at Nat. Comm.

1) In Figure 1, it is shown that TALEs specifically target subtelomeres containing the 29 bp repeat consensus and not other subtelomeres nor non-telomeric loci. More importantly, TERRA

expression is downregulated exclusively at the aforementioned telomeres. However, it is not shown whether expression of TALEs affects the expression of other genes that might have an effect on ALT telomeres i.e. recombination factors, DNA polymerase delta subunits etc. I would recommend an RNA seq experiment to ensure that known factors for ALT maintenance are not affected (e.g. FANCM).....or at the least western blots for as many known ALT maintenance factors as possible.

We understand the concern of this Reviewer and following his/her suggestion we performed RNAseq experiments using total RNA from nls3 and sid4 cells treated with dox for 72 hours or left untreated. Comparative analysis of dox-treated versus untreated samples identified significant differential expression of 3063 transcripts in sid4 cells and only 15 in nls3 cells (data are shown in Fig. 6a and supplementary Table S2). Transcript level alterations in sid4 cells were overall moderate, with only 226 and 236 transcripts showing a log₂ fold change of at least 1 or -1, respectively (Table S2). We conclude that TERRA transcription suppression is associated with mild yet significant transcriptomic changes in human ALT cells. This is consistent with previous work from the Lee laboratory showing that TERRA depletion in mouse ES cells caused gene expression alterations⁶ (this is now mentioned in the text, lanes 230-242).

Careful inspection of RNAseq data identified 11 protein coding transcripts (downregulated: ORC1, RAD51AP1, BRCA2, GEN1, ORC6, BRCA1, HROB, BLM, EXO1, FANCD2; upregulated: LIG4) whose altered expression could alleviate telomere instability and/or ALT activity (Fig. 6a). The differential expression of those transcripts was borderline significant; nonetheless, we quantified the corresponding protein levels by western blotting in nls and sid cells treated with dox for 0, 24 and 72 hours. Although fluctuations in protein levels could be detected for some factors, they likely derived from a general, short-term response to dox, as they were observed in all cell lines and only at 24 hours of treatment (data shown in Fig. 6c). This confirms that the effects exerted by T-TALE expression on telomere stability and ALT activity derive from TERRA transcription suppression and not from deregulated off-target transcripts (this is now discussed in the text, lanes 249-261).

We also performed Gene Ontology (GO) analysis of differentially expressed protein coding transcripts and identified significant alterations in 196 GO terms traceable to 12 main biological processes (data shown in Fig. 6b and Table S3). Interestingly, we observed significant upregulation of transcripts involved in vacuole organization and autophagy, and downregulation of transcripts involved in apoptotic cell death (data shown in Fig. S7 and Table S3). We find this observation of interest because it suggests that selective killing of ALT cancer cells could potentially be achieved through TERRA transcription suppression. We have mentioned this in the text (lanes 242-248 and 363-368).

References

1. Graf, M. *et al.* Telomere Length Determines TERRA and R-Loop Regulation through the Cell Cycle. *Cell* **170**, 72-85 e14 (2017).
2. Moravec, M. *et al.* TERRA promotes telomerase-mediated telomere elongation in *Schizosaccharomyces pombe*. *EMBO Rep* **17**, 999-1012 (2016).
3. Arnoult, N., Van Beneden, A. & Decottignies, A. Telomere length regulates TERRA levels through increased trimethylation of telomeric H3K9 and HP1alpha. *Nat Struct Mol Biol* **19**, 948-56 (2012).
4. Arora, R. *et al.* RNaseH1 regulates TERRA-telomeric DNA hybrids and telomere maintenance in ALT tumour cells. *Nat Commun* **5**, 5220 (2014).

5. Dilley, R.L. *et al.* Break-induced telomere synthesis underlies alternative telomere maintenance. *Nature* **539**, 54-58 (2016).
6. Chu, H.P. *et al.* TERRA RNA Antagonizes ATRX and Protects Telomeres. *Cell* **170**, 86-101 e16 (2017).

REVIEWERS' COMMENTS

Reviewer #1 (Remarks to the Author):

The reviewers addressed in full all my (minor) points. This is a very strong manuscript that will be of great interest to the telomere biology community.

Reviewer #2 (Remarks to the Author):

I find the revised manuscript improved in many key respects. The authors have carefully addressed most criticisms and therefore, on balance, I think that the study both represents a solid contribution of general interest and significance, and carefully discusses advances and limitations inherent in the work.

I am puzzled by the failure to see an enrichment in the new ChIP data provided in Figure 2. Telomeric ChIP is normally carried out with a dot-blot approach rather than Q-PCR so, at least in principle, the explanation provided (that the protein association might not extend all the way to the subtelomere - even though the location of the amplicon is very close to the telomere repeat array) can in principle explain the result. However, it would have been nice to provide some controls, for example using a TRF2 dominant negative or FoKI-TRF1 construct. Similarly, the telomere length analysis would have been best carried out using a telomere-specific assay, STELA, which would have the important significant benefit of zooming on critically shortened telomeres. Still, I think that the limitations in the interpretation of the results (cis vs trans effect of TERRA) are properly highlighted and there is enough here to justify publication.

Point-by-point response to the Reviewers' comments.

Reviewer #1 (Remarks to the Author):

The reviewers addressed in full all my (minor) points. This is a very strong manuscript that will be of great interest to the telomere biology community.

We would like to thank this Reviewer for his/her support.

Reviewer #2 (Remarks to the Author):

I find the revised manuscript improved in many key respects. The authors have carefully addressed most criticisms and therefore, on balance, I think that the study both represents a solid contribution of general interest and significance, and carefully discusses advances and limitations inherent in the work.

We would like to thank this Reviewer for his/her support.

I am puzzled by the failure to see an enrichment in the new ChIP data provided in Figure 2. Telomeric ChIP is normally carried out with a dot-blot approach rather than Q-PCR so, at least in principle, the explanation provided (that the protein association might not extend all the way to the subtelomere - even though the location of the amplicon is very close to the telomere repeat array) can in principle explain the result. However, it would have been nice to provide some controls, for example using a TRF2 dominant negative or FoKI-TRF1 construct. Similarly, the telomere length analysis would have been best carried out using a telomere-specific assay, STELA, which would have the important significant benefit of zooming on critically shortened telomeres. Still, I think that the limitations in the interpretation of the results (cis vs trans effect of TERRA) are properly highlighted and there is enough here to justify publication.

We agree that a positive control would have been useful, however, at this point we are not sure which control would be the most appropriate. With the suggested ones (TRF2 dominant negative and FokI-TRF1) we expect to detect increased \$\gamma\$ H2AX also by qPCR with subtelomeric oligonucleotides; indeed, both transgenes would generate DNA damage at the very first telomeric repeats. On the contrary, we do not know where the damage induced by TERRA transcription accumulates along the telomeric tract. Also, we agree that STELA could have been used to measure telomere length and we will consider this for future experiments, thank you very much for the suggestions.